# PM₂.₅ and O₃ in an Enclosed Basin, the Guanzhong Basin of Northern China: Insights into Distributions, Appointment Sources, and Transport Pathways

**Xiaofei Li** [1,2,3,4,*], **Jingning Guo** [1], **Xuequan Wan** [1], **Zhen Yang** [1], **Lekhendra Tripathee** [4], **Feng Yu** [1], **Rui Zhang** [1], **Wen Yang** [1] and **Qiyuan Wang** [2,3]

[1] School of Environmental Science and Engineering, Shaanxi University of Science and Technology, Xi'an 710021, China; 210311031@sust.edu.cn (J.G.); 202103040420@sust.edu.cn (X.W.); 210311009@sust.edu.cn (R.Z.)

[2] National Observation and Research Station of Regional Ecological Environment Change and Comprehensive Management in the Guanzhong Plain, Xi'an 710061, China

[3] Key Laboratory of Aerosol Chemistry and Physics, State Key Laboratory of Loess and Quaternary Geology, Institute of Earth Environment, Chinese Academy of Sciences, Xi'an 710061, China

[4] State Key Laboratory of Cryospheric Science, Northwest Institute of Eco-Environment and Resources, Chinese Academy of Sciences, Lanzhou 730000, China

* Correspondence: lixiaofei@ieecas.cn

**Abstract:** Aerosol samples (PM₂.₅) were collected in Xi'an (XN) from 11 August to 11 September 2021 and in Qinling (QL) from 14 July to 24 August 2021, respectively. In addition, ozone (O₃) data were collected in order to investigate the characteristics and source areas of PM₂.₅ and O₃ in the Guanzhong Basin (GB). The concentrations of PM₂.₅, organic carbon (OC), and elemental carbon (EC) in XN (53.40 ± 17.42, 4.61 ± 2.41, and 0.78 ± 0.60 µg m⁻³, respectively) were higher than those in QL (27.57 ± 8.27, 4.23 ± 1.37, and 0.67 ± 0.53 µg m⁻³, respectively) in summer. Total water-soluble ions (TWSIIs) accounted for 19.40% and 39.37% of the PM₂.₅ concentrations in XN and QL, respectively. O₃ concentrations in summer were 102.44 ± 35.08 µg m⁻³ and 47.95 ± 21.63 µg m⁻³ in XN and QL, respectively, and they showed a significant correlation with Oₓ. The positive matrix factorization (PMF) model identified three main sources in XN and QL, including coal combustion source (COB), secondary aerosol (SA), and dust sources (DUSs). The potential source contribution function (PSCF) and a concentration weight trajectory (CWT) model with back-trajectory analysis showed that Inner Mongolia, the interior of Shaanxi, and nearby areas to the southwest were the sources and source areas of carbonaceous matter in XN and QL. The results of this study can contribute to the development of prevention and control policies and guidelines for PM₂.₅ and O₃ in the GB. Furthermore, long-term and sustainable measuring and monitoring of PM₂.₅ and O₃ are necessary, which is of great significance for studying climate change and the sustainable development of the environment.

**Keywords:** carbonaceous matter; water-soluble ions; O₃; potential sources; sustainability; Guanzhong Basin

## 1. Introduction

Aerosol of PM₂.₅ (particulate matter with aerodynamic diameter less than or equal to 2.5 µm) and its formation process and sources are very complex [1]. As a result, it has an important impact on the climate, the environment, and human health [2]. The main components include carbon matter, water-soluble ions (WSIIs), and metallic elements. It can easily enter the human body through the respiratory system and further affect physical health. Studies indicate that PM₂.₅ has an impact on the respiratory system, causes cognitive impairment in the elderly, increases the risk of cancer, and causes arrhythmia; fur-

thermore, it can lead to impaired vasoconstriction and accelerated arteriosclerosis [3]. Carbonaceous aerosols are an important part of PM$_{2.5}$, accounting for about 20~50% of urban atmospheric aerosols in China [4]. These mainly include elemental carbon (EC) produced through the direct emission of pollutants and organic carbon (OC) from more complex sources [5]. OC is further divided into water-soluble organic carbon (WSOC) and water-insoluble organic carbon (WIOC), which have a significant impact on the Earth's radiation balance, global climate change, and human health [2,6]. EC has a strong capacity for absorbing visible light and solar radiation [7] and absorbs and warms the atmosphere, making it the second largest greenhouse gas after CO$_2$ [8]. OC is usually considered a scattering component without a warming effect. In addition, WSIIs account for about 30~80% of PM$_{2.5}$, making them an important component. They are hygroscopic and soluble [9,10] and play an important role in the optical properties of particulate matter [11].

Relevant studies have found that surface ozone (O$_3$) has a warming effect as a greenhouse gas; at the same time, it has a significant impact on human health and the ecosystem due to its strong oxidizing capability. More importantly, increasing concentrations of O$_3$ accelerate the formation of PM$_{2.5}$ and other pollutants and further affect the ambient air quality [12–16]. Data show that in recent years, the concentration of O$_3$ in China has shown a clear upward trend [12,17,18]. O$_3$ pollution has become another environmental problem, in addition to PM$_{2.5}$ pollution, that needs to be solved urgently in China. However, in recent years, PM$_{2.5}$ pollution has received widespread attention and has been effectively controlled [19,20], but O$_3$ has been on the rise in most parts of China [21]. Therefore, an exploration of the characteristics of O$_3$ and the compound pollution of PM$_{2.5}$ and O$_3$ is necessary. Through previous research, O$_3$ pollution has been found in most parts of China in recent studies [18,21–24]. However, research on PM$_{2.5}$ and O$_3$ in the Guanzhong Basin (GB) is relatively limited. As a mega city in the GB, Xi'an (XN) has severe air pollution problems. Qinling (QL) is an ecologically protected area in the GB that is less polluted by human activities; therefore, a more comprehensive perspective is needed to study the characteristics of PM$_{2.5}$ and O$_3$ in the GB. PM$_{2.5}$ and O$_3$ are not only affected by local human activities but also highly susceptible to remote source transport due to the unique terrain conditions of the GB and the influence of the summer southeast monsoon. However, there are relatively few studies on the source and source areas of PM$_{2.5}$ and O$_3$ in the GB during summer. Therefore, studying the sources of PM$_{2.5}$ and O$_3$ in the GB in the summer is of great significance for understanding the transport law of atmospheric pollutants and the prevention and control of atmospheric pollutants in this area.

This study reported the characteristics of O$_3$, the chemical composition of PM$_{2.5}$, and the links between them in the GB in summer. Secondly, it quantitatively assessed the source and source areas of PM$_{2.5}$, its components, and O$_3$ in the GB. To this end, PM$_{2.5}$ samples were collected in XN from 11 August to 11 September 2021 and in QL from 14 July to 24 August 2021 (data were missing for the periods 17 July–23 July, 25 July, 7 August–9 August, 12 August, and 13 August 2021 due to unexpected events). In addition, the characteristics of O$_3$ were analyzed using O$_3$ data downloaded from the National Climatic Data Center (NCDC) from 11 August to 11 September 2021 in XN and from 14 July to 24 August 2021 in QL. At the same time, O$_3$ and PM$_{2.5}$ data for XN and QL during winter were collected from NCDC (from 1 January to 31 January 2021) for comparison in order to more clearly explain the characteristics of summer O$_3$ characteristics in the GB. Finally, the backward trajectory clustering techniques, the potential source contribution function (PSCF) model, the concentration weighted trajectory (CWT) method, and the positive matrix factorization (PMF) model were used to evaluate the transport pathways and sources of PM$_{2.5}$ and O$_3$ in XN and QL. This study will provide a better understanding of the pollution characteristics of PM$_{2.5}$ and O$_3$ in the GB. The results of this study will also contribute to formulating prevention and control policies and guidelines for PM$_{2.5}$ and O$_3$ in China.

## 2. Materials and Methods

## 2.1. Sample Site

XN is located in the GB city cluster in Northwest China, and its unique geographical location conditions are favorable for the formation and accumulation of severe air pollutants (Figure 1). XN has a warm, temperate, semi-humid continental monsoon climate, with distinct cold, warm, dry, and wet seasons. Summer is hot and rainy, with prevailing southeast winds; winter is dry and cold, with prevailing northwest winds. As of 2020, it is a mega city that covers 10,752 km² with a population of around 130 million. The city is located at an altitude of around 400 m. The sampling site was located on the roof of a building in XN (108.97° E, 34.37° N; ~30 above ground level). The building is adjacent to the highway and surrounded by teaching areas and student living areas. There are no large industrial emission sources around the sampling site.

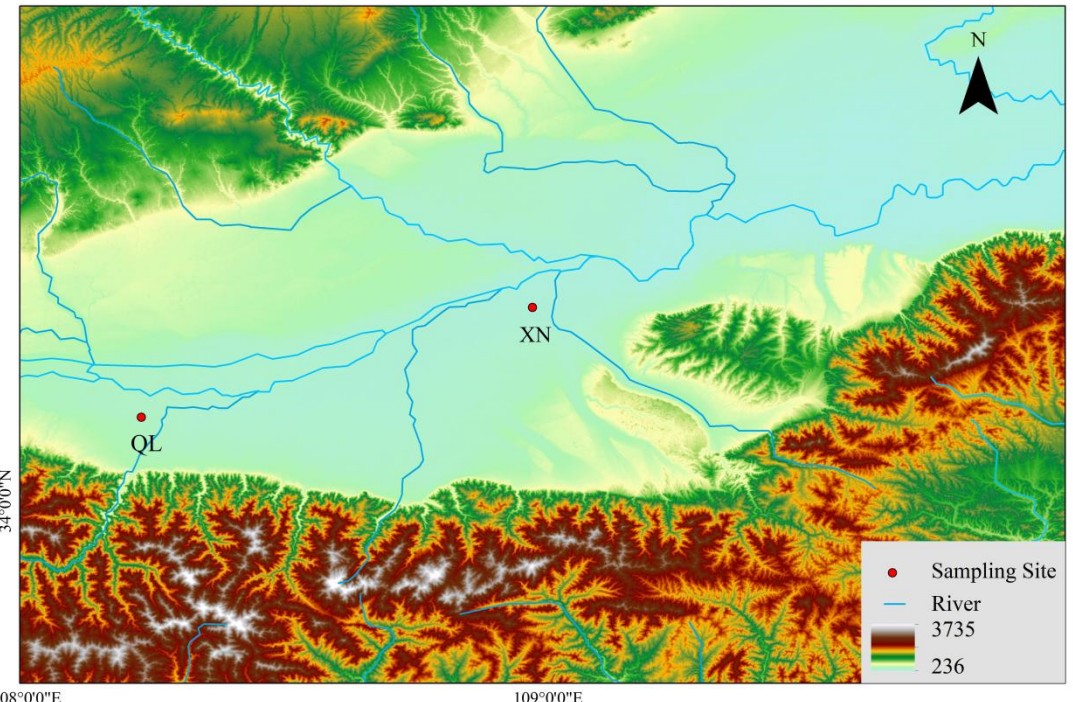

**Figure 1.** Location map of sampling points.

QL is located in the south of the GB; it is an important ecological protection area in the region and has important ecological functions (Figure 1). The city is located at an average altitude of 2400 m. The sampling site in this city was located in the suburbs far from the city (108.22° E, 34.16° N). The surrounding vegetation is abundant, and human pollution is relatively low.

## 2.2. Sample Collection

$PM_{2.5}$ was collected continuously using a medium-flow sampler (HC-1010, Qingdao, China). A total of 32 and 29 valid samples were obtained in XN and QL, respectively, during summer (except during extreme weather and unexpected conditions), and blank samples were also collected as controls. Sampling details are described in Text S1. At the same time, the data of $PM_{2.5}$ for XN and QL during winter (from 1 January to 31 January 2021) were obtained directly from NCDC. Meteorological parameters, including wind direction, wind speed, temperature, and humidity data, were downloaded from NCDC. Wind rose diagrams for XN and QL in summer are shown in Figure S1.

### 2.3. *Chemical Analyses*

2.3.1. Carbon Matter

OC and EC were measured using a DRI Model 2015 carbon analyzer, following a thermal/optical carbon analysis method based on the IMPROVE-A protocol [25]. The detailed process of carbon matter determination is presented in Text S2.

The EC tracer method was used to estimate the concentrations of primary organic carbon (POC) and secondary organic carbon (SOC), and the calculation formula is detailed in Text S3.

2.3.2. Water-Soluble Ions

WSIIs ($NH_4^+$, $K^+$, $Na^+$, $Ca^{2+}$, $Mg^{2+}$, $Cl^-$, $F^-$, $SO_4^{2-}$, $NO_2^-$, and $NO_3^-$) were measured using an ion chromatograph. The pretreatment process for WSIIs is described in detail in Text S4. The calculations of the acidity, alkalinity, sulfur oxidation rates (SOR), and nitrogen oxidation rates (NOR) of WSIIs are detailed in Text S5. The data of $SO_2$ and $NO_2$ used for SOR and NOR calculations were obtained from NCDC.

### 2.4. *Source Analysis*

2.4.1. PMF Model

The daily concentration data for $PM_{2.5}$, OC, EC, and 10 WSIIs ($NH_4^+$, $K^+$, $Na^+$, $Ca^{2+}$, $Mg^{2+}$, $Cl^-$, $F^-$, $SO_4^{2-}$, $NO_2^-$, and $NO_3^-$) were input into the PMF model to identify their sources. The details of the PMF model are presented in Text S6.

2.4.2. 48-H Backward Trajectory

Backward trajectory clustering, PSCF, and CWT were calculated using the GIS-based software TrajStat [26]. Details of the PSCF and CWT are described in Text S7.

$W_{ij}$ was introduced to reduce the uncertainty of PSCF and CWT [27]. The values of $W_{ij}$ are described in detail in Text S8.

## 3. Results and Discussion

### 3.1. *Characteristics of PM2.5 and Major Components*

3.1.1. $PM_{2.5}$, OC, and EC

Table 1 provides a summary of the concentrations of $PM_{2.5}$ for the two sites in summer. The concentration of $PM_{2.5}$ in XN ($53.40 \pm 17.42$ µg m$^{-3}$) was higher than that in QL ($27.57 \pm 8.27$ µg m$^{-3}$); this showed that QL had excellent air quality in summer as a significant natural ecological reserve. Compared to previous studies [28,29], it can be seen that the concentrations of $PM_{2.5}$ in XN and QL in summer were significantly lower than in winter.

**Table 1.** The average concentration of $PM_{2.5}$ components measured in this study.

| | XN | | | QL | | |
|---|---|---|---|---|---|---|
| | **Avg** | **Max** | **Min** | **Avg** | **Max** | **Min** |
| $PM_{2.5}$ (µg m$^{-3}$) | $53.40 \pm 17.42$ | 82.64 | 13.89 | $27.57 \pm 8.27$ | 48.10 | 11.38 |
| OC (µg m$^{-3}$) | $4.61 \pm 2.41$ | 9.94 | 0.90 | $4.23 \pm 1.37$ | 7.73 | 1.35 |
| EC (µg m$^{-3}$) | $0.78 \pm 0.60$ | 2.15 | 0.10 | $0.67 \pm 0.53$ | 2.03 | 0.03 |
| OC/EC | $9.84 \pm 6.95$ | 24.53 | 2.02 | $10.00 \pm 6.99$ | 28.12 | 3.18 |
| POC (µg m$^{-3}$) | $1.54 \pm 1.23$ | 4.32 | 0.16 | $2.19 \pm 1.65$ | 6.46 | 0.22 |
| SOC (µg m$^{-3}$) | $3.32 \pm 1.65$ | 6.18 | 0.90 | $2.35 \pm 1.05$ | 4.72 | 0.18 |

The concentrations of OC and EC in XN ($4.61 \pm 2.41$ and $0.78 \pm 0.60$ µg m$^{-3}$, respectively) were higher than in QL ($4.23 \pm 1.37$ and $0.67 \pm 0.53$ µg m$^{-3}$, respectively) (Table 1), but they were significantly lower in summer compared with winter [29]. In summer, OC

and EC concentrations in XN were lower than those in Shijiazhuang [30] and in Guangzhou [31], but they were higher than that in the Tibetan Plateau [32–34]. At the same time, the variance in OC was greater than that in EC in XN and QL, which indicated that the emission of EC was more stable. Generally, EC is generated through the direct discharge of pollutants, while the sources of OC are more complex, including POC produced through direct emission and SOC produced through the photochemical reaction of organic matter [35]. This can lead to greater variability in OC particles compared with EC. The correlation of OC and EC can reflect whether the source is consistent [36]. The correlation between OC and EC was weak in XN ($R^2 = 0.45$, $p < 0.01$) and QL ($R^2 = 0.53$, $p < 0.01$) in summer, which revealed that the OC and EC components in the GB were emissions from different sources. Conversely, a stronger correlation was observed between OC and EC in XN ($R^2 = 0.76$, $p < 0.01$) and QL ($R^2 = 0.95$, $p < 0.01$) in winter, which may have been caused by synchronized effects of residential heating in winter and motor vehicle exhaust emissions (Figure S2). In conclusion, the correlation between OC and EC was weaker in summer and stronger in winter, with similar conclusions reported by Ji et al. (2016) in summer in Beijing.

The value of OC/EC is an important index of reaction pollution sources [35]. In summer, OC/EC values were 9.84 ± 6.95 and 10.00 ± 6.99 in XN and QL, respectively (Table 1). When the OC/EC value is greater than 2, it indicates SOC generation [35]. POC is produced through the direct emission of pollutants, and SOC is formed by gaseous organic matter through atmospheric photochemical and other reactions [37]. There was more SOC generation in the GB during summer, perhaps because the atmospheric photochemical reactivity was stronger during this period, which was therefore more conducive to the formation of SOC. Meanwhile, the SOC concentration in summer was significantly lower than that in winter. One study showed that a 10 °C temperature increase results in an 18% reduction in SOC concentration due to lower temperatures benefiting the adsorption of organic compounds on existing particles [37]. However, the formation of SOC is very complex, and the mechanism of its formation needs further investigation.

3.1.2. Water-Soluble Ions

The concentrations of the 10 WSIIs are shown in Table S1. WSIIs accounted for 19.40% and 39.37% of the $PM_{2.5}$ mass in XN and QL, respectively. The proportion of WSIIs in $PM_{2.5}$ in XN was significantly lower compared with 2006 (42.0 ± 8.9%), 2008 (44.6 ± 12.4%), and 2010 (35.6 ± 10.2%) [38]. The proportion of WSIIs in $PM_{2.5}$ in XN and QL was significantly lower than that in Qingdao (62%) [39]. $NH_4^+$, $SO_4^{2-}$, and $NO_3^-$ were the most abundant in XN, accounting for 22.75%, 21.89%, and 17.00% of the total ion mass, respectively; $Na^+$, $SO_4^{2-}$, and $NH_4^+$ were the most abundant in QL, accounting for 45.56%, 28.09%, and 9.28% of the total ions mass, respectively. Atmospheric $H_2SO_4$ was produced through the gas phase reaction of HO and $SO_2$ or through the liquid phase reaction of $H_2O_2$ or $O_3$ and $SO_2$. The higher $SO_4^{2-}$ concentration might be due to the higher conversion efficiency of $SO_2$ in summer. It has been shown that anions such as $F^-$, $Cl^-$, $NO_2^-$, $SO_4^{2-}$, and $NO_3^-$ can increase the acidity of particulate matter, while cations such as $Na^+$, $NH_4^+$, $K^+$, $Mg^{2+}$, and $Ca^{2+}$ can increase its alkalinity [40]. The acidity and alkalinity of particulate matter also affect the pH of atmospheric precipitation, which not only leads to the acidification of precipitation but also plays a neutral role in the acidity of regional precipitation. Usually, the acidity of particulate matter is judged based on the ion charge balance [40]. In summer, the values of AE/CE in XN and QL were 0.41 and 0.28, respectively, which were less than 1, indicating that the atmospheric particulate matter in the GB was mostly alkaline. It is speculated that the alkaline appearance of atmospheric particles in summer may be caused by long-distance transmission of atmospheric particulate matter and the neutralization of local emissions of alkaline particles.

The correlation analysis of WSIIs is shown in Table S2 and Figure 2. The correlation coefficient $R^2$ of $K^+$ and $Cl^-$ was 0.321 in QL ($p < 0.01$) (Figure 2e), which was stronger than that of XN. This indicates that $K^+$ and $Cl^-$ had the same sources, and $K^+$ had multiple

sources, mainly from soil dust, that is, biomass burning and vegetation [41]. The correlation between $Ca^{2+}$ and $Mg^{2+}$ is shown in Figure 2f. The correlation coefficients $R^2$ of $Ca^{2+}$ and $Mg^{2+}$ were 0.460 and 0.931 in XN and QL ($p < 0.01$), respectively. The correlation between $Ca^{2+}$ and $Mg^{2+}$ was significantly stronger in QL than in XN, indicating that while the two sampling sites were affected by road dust emissions, more importantly, QL was more significantly affected by road dust emissions [42]. At the same time, there was a significant correlation between $NH_4^+$ and $SO_4^{2-}$ ($R^2 = 0.709$, $p < 0.01$) (Figure 2b) and $NH_4^+$ and $NO_3^-$ ($R^2 = 0.546$, $p < 0.01$) (Figure 2c) in QL, indicating that there might be three formations of ammonium in the atmosphere, including $NH_4NO_3$, $(NH)_2SO_4$, and $NH_4HSO_4$. However, the presence of ammonium in the atmosphere is relatively multiple, and further research is needed to accurately assess its presence.

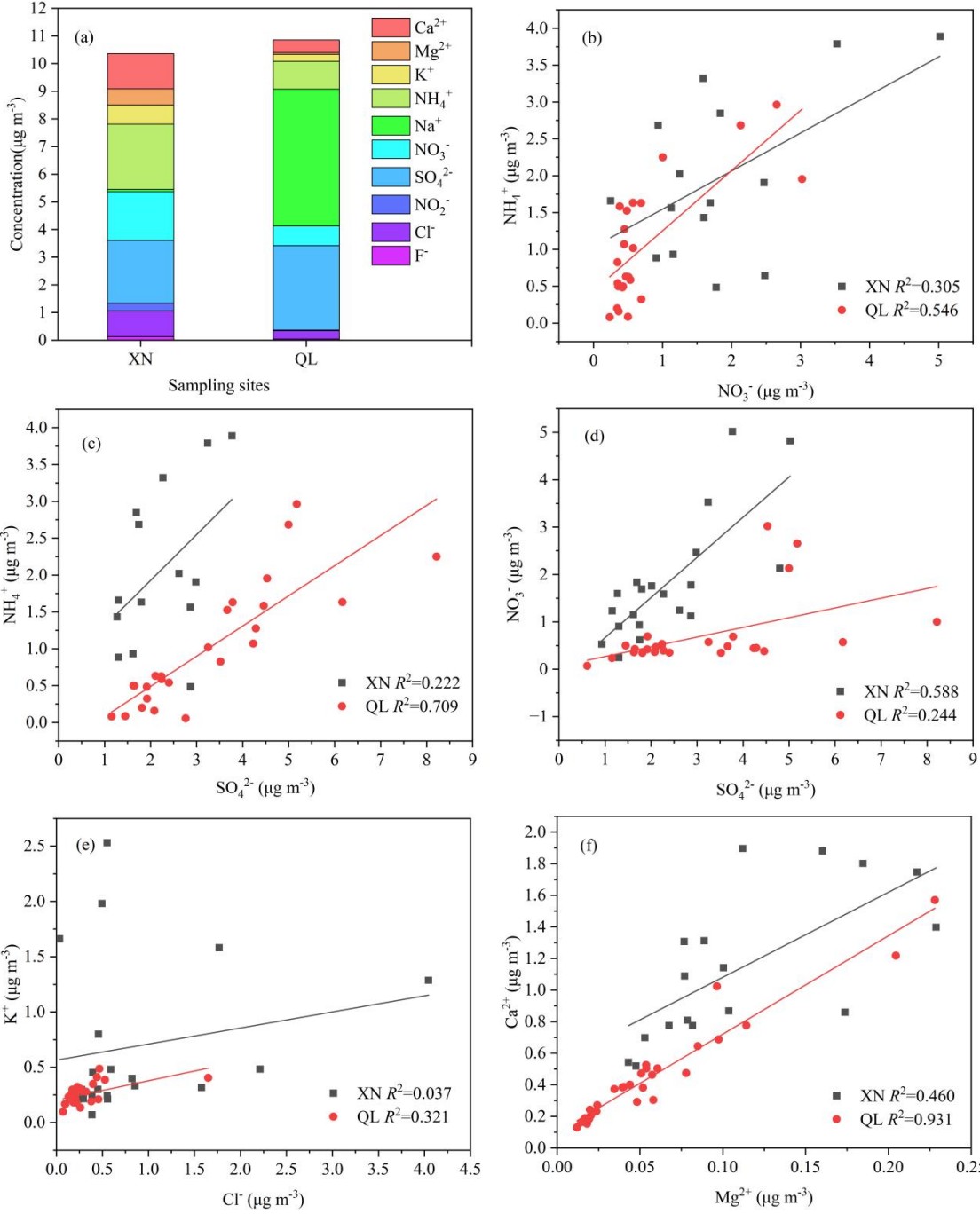

**Figure 2.** WSIIs concentrations and correlations. (**a**) Concentration of ions; (**b**–**f**) correlation of the main ions.

$SO_4^{2-}$ is generated through the complex oxidation process of $SO_2$ from coal emissions in the atmosphere, and $NO_3^-$ is mainly generated from the $NO_2$ released from automobile exhaust, industrial, and biomass combustion emissions [43]. The weight of the contribution of fixed sources (such as coal combustion) and mobile sources (such as automobile exhaust) can be roughly judged using the $NO_3^-/SO_4^{2-}$ ratio in atmospheric particulate matter [44]. When the $NO_3^-/SO_4^{2-}$ ratio exceeds 1, the contribution of the mobile source is greater than that of the fixed source; otherwise, the contribution of the fixed source is larger. The values of $NO_3^-/SO_4^{2-}$ were 0.77 and 0.23 in XN and QL, respectively, both less than 1; however, the $NO_3^-/SO_4^{2-}$ ratio was closer to 1 in XN, indicating that the contribution of mobile sources in this city was not negligible.

SOR and NOR indicate the degree of conversion of $SO_2$ and $NO_2$ to the secondary ions $SO_4^{2-}$ and $NO_3^-$, respectively. When the SOR is greater than 0.25 and NOR is greater than 0.1, the photochemical oxidation of $SO_2$ and $NO_2$ occurs in the atmosphere [44]. The SOR value was greater than 0.25 in QL (0.516), while it was below 0.25 in XN (0.200), indicating that photochemical oxidation occurs in summer in QL, as high temperatures promote the oxidation of gaseous $SO_2$ to $SO_4^{2-}$. The NOR value was lower than 0.1 in XN but higher than 0.1 in QL, indicating that the degree of secondary conversion of $NO_2$ was higher in QL. The NOR value was significantly lower than SOR in summer in the GB. In the summer, $NO_2$ could be converted into $NO_3^-$ and combine with $NH_3$ to become nitrate, while the higher temperatures in summer facilitate the decomposition of $NH_4NO_3$, thus decreasing the concentration of $NO_3^-$ [45]. However, the mechanisms of nitrate and its conversion were affected by many factors. More thorough investigations and data are required to elucidate the conversion mechanism of nitrate.

*3.2. Characteristics of $O_3$*

$O_3$ is an important greenhouse gas [46]. In this study, the concentrations of $O_3$ were $102.44 \pm 35.08$ μg m$^{-3}$ and $44.48 \pm 15.15$ μg m$^{-3}$ in XN during summer and winter, respectively (Figure 3). The concentrations of $O_3$ were $47.95 \pm 21.63$ μg m$^{-3}$ and $25.87 \pm 5.74$ μg m$^{-3}$ in QL during summer and winter, respectively (Figure S3). The $O_3$ concentration showed significant seasonal variation and is maintained at higher levels in summer in the GB. $O_3$ is generated through a series of complex reactions of VOCs and $NO_x$ emitted from anthropogenic, biogenic, and biomass burning sources. In China, $O_3$ is more susceptible to the influence of industrial sources in summer. At the same time, it is also affected by biogenic emissions [47]. For example, it was found in one study that biogenic emissions contributed 8.2 ppb to the concentration of $O_3$ in XN [48]. Sunlight lasts for a long time in this region in summer, with strong solar radiation and thus a strong photochemical reaction capacity. When the solar radiation is stronger, the efficiency of NO in consuming $O_3$ is reduced, leading to the accumulation of more $O_3$ in the atmosphere in summer [49]. The concentration of $O_3$ is positively correlated with temperature and the intensity of solar radiation; intense solar radiation and high temperature can promote photochemical reaction processes, and under strong light and high-temperature conditions, $NO_2$ will undergo photolysis to form $O_3$ [50]. The reaction process of photochemical pollution is represented by formulae (1), (2), and (3). At the $O_3$ accumulation stage, VOCs and OH react to produce a large number of peroxyl radicals (e.g., $HO_2$ and $RO_2$), which can convert NO into $NO_2$. As solar radiation gradually increases, $NO_2$ is photodegraded into $O_3$.

$$OH + VOCs + O_2 - HO_2 + RO_2 \tag{1}$$

$$RO_2 + NO - RO + NO_2 \tag{2}$$

$$NO_2 + hv + O_2 - NO + O_3 \tag{3}$$

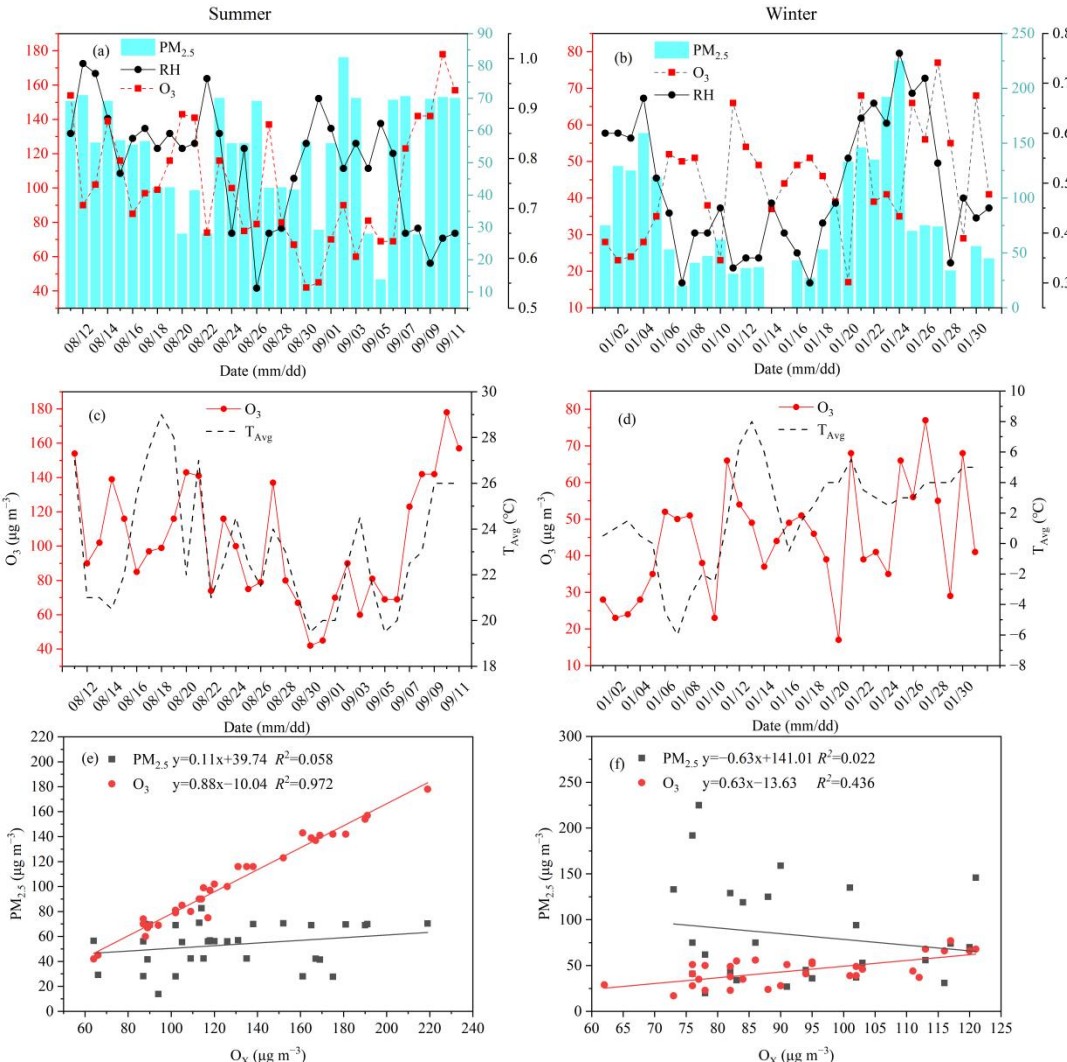

**Figure 3.** The relationship between $O_3$ and $PM_{2.5}$ and meteorological factors in XN; (**a**,**b**) the distribution of $PM_{2.5}$, RH, and $O_3$ in summer and winter, respectively; (**c**,**d**) the distribution of $O_3$ and $T_{Avg}$ in summer and winter, respectively; (**e**,**f**) the correlation between $PM_{2.5}$ and $O_3$ in summer and winter.

However, compared with previous studies, the concentration of $O_3$ during summer in XN was lower than that in the North China Plain (117.89–154.20 μg m⁻³) [51] but higher than that in Beijing (91.26 μg m⁻³) [24]. In general, a high level of $O_3$ pollution occurred in XN during summer.

The atmospheric total oxidant $O_x$ ($O_x = O_3 + NO_2$) is an important indicator for identifying atmospheric oxidizing ability [52]. The larger the value, the stronger the atmospheric oxidation capacity. The relationship between $PM_{2.5}$ and $O_3$ is very complex; $O_3$ contamination largely depends on the concentration and chemical components of $PM_{2.5}$. During summer, the correlation coefficients $R^2$ of $O_x$ and $PM_{2.5}$ were 0.058 and 0.143 in XN and QL ($p < 0.01$), respectively, while the correlation coefficients $R^2$ of $O_x$ and $O_3$ were 0.972 and 0.997 in XN and QL ($p < 0.01$) (Figure 3, Figure S3), respectively, indicating that atmospheric oxidation capacity is an important cause of $O_3$ pollution, and it has become the main cause of $O_3$ pollution in summer. The higher the $O_3$ concentration, the stronger the correlation.

Sulfate, nitrate, and carbonaceous aerosols in $PM_{2.5}$ can directly scatter or absorb solar radiation, indirectly changing the optical properties and life cycle of clouds and affecting

the intensity of ultraviolet radiation intensity, thus affecting the photolysis rate and generation of $O_3$ [53]. A high concentration of $O_3$ means that there is strong photochemical reactivity in the atmosphere and promotes the formation of SOC. For example, when the photochemical reaction in the atmosphere is strong, $SO_2$ will be photochemically oxidized to $H_2SO_4$, and when the $O_3$ concentration is high, it will promote the conversion of $NO_2$ to $HNO_3$. In this study, $PM_{2.5}$ and $O_3$ were positively correlated in summer, and their concentrations synchronously increased or decreased overall. The main reason for this is that the temperature is high and the radiation is strong in summer, and the atmospheric particles enhance the downward radiation flux of the sun through the scattering of aerosol particles, which is conducive to the generation of $O_3$. In addition, a high $O_3$ concentration, strong light, and enhanced atmospheric oxidation further promote the generation of SOC [54].

China is influenced by the Asian monsoon, and seasonal variations in $O_3$ may be influenced by the winter and summer monsoon characteristics. According to previous studies, the characteristics of the Asian monsoon are usually described using monsoon indices such as pressure, temperature, wind direction, and humidity [55]. The main meteorological factors affecting $O_3$ are temperature, sunshine, relative humidity, and wind speed [56]. This study takes XN as an example to analyze the effects of temperature and relative humidity on $O_3$ formation (Figure 3a–d). In summer and winter, there was an opposite trend of variation between $O_3$ and relative humidity, and $O_3$ decreased with increasing relative humidity. However, it is worth noting that in summer, when the relative humidity was more than 50%, $O_3$ and relative humidity showed an opposite trend of variation. The trend of variation between $O_3$ concentration and relative humidity when the relative humidity is less than 50% requires more discussion. This phenomenon was discussed in more detail by Song and Hao (2022). In summer, the average temperature was greater than 18 °C. Figure 3c shows roughly the same trend of variations for the concentration of $O_3$ and the temperature of the atmosphere, but this trend was not significant in winter. The change in the solar radiation intensity is reflected by the temperature. This indicates that when the temperature was high in summer, the concentration of $O_3$ was significantly affected by the temperature of the atmosphere, and the concentration of $O_3$ increased with the increase in the temperature.

### 3.3. Source of $O_3$, $PM_{2.5}$, and Major Components

3.3.1. PMF Model

The PMF model showed three factors for XN and QL during the sampling period (Figures 4, S4 and S5). The detailed data are listed in Table S3. The sources of WSIIs and carbon matter in XN and QL are defined as dust sources (DUSs), secondary aerosol (SA), and coal combustion source (COB) (Figure 4). As shown in Figure S4, factor 1 was defined as DUSs in XN due to the high loading of $Mg^{2+}$ and $Ca^{2+}$; factor 2 was defined as SA due to the high loading of $SO_4^{2-}$ and $NO_3^-$ [57,58]; and factor 3 was defined as COB due to the high loading of EC and OC. As shown in Figure S5, in QL, factor 1 was defined as DUSs; factor 2 was defined as SA due to the high loading of $NH_4^+$, $SO_4^{2-}$, and $NO_3^-$; and factor 3 was defined as COB. The contribution rate of the pollution sources of each component in $PM_{2.5}$ is shown in Figure 4 according to the above analysis results. In XN, the contribution percentages of the pollutants were COB (38%), DUSs (34.2%), and SA (27.9%); In QL, the contribution percentages of the pollutants were SA (40.4%), DUSs (33.6%), and COB (26%).

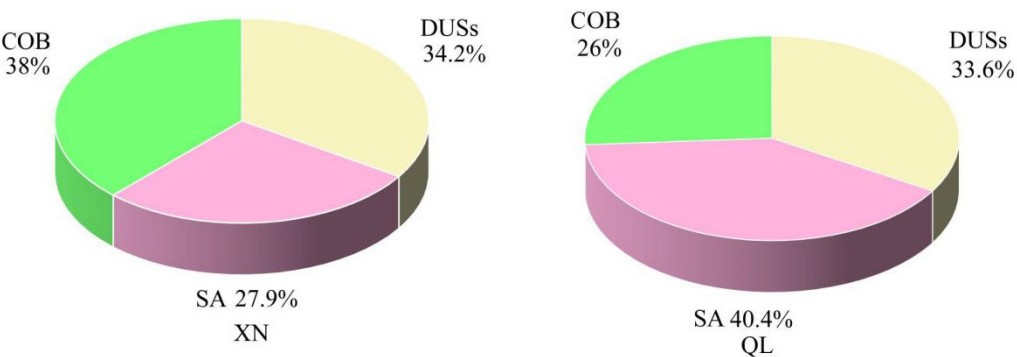

**Figure 4.** The percentage of contribution of the three identified sources in XN and QL.

In conclusion, PM2.5 in XN and QL was affected by COB, DUSs, and SA in the summer. This is partly due to artificial activities such as straw burning and cooking in summer and occasional forest fires in QL. The sampling station in XN is close to a highway, which may be more easily affected by road dust. In addition, the strong solar radiation and high temperatures in summer are more likely to lead to the generation of SA. In addition, long-distance transport may also affect the atmospheric environments of XN and QL.

### 3.3.2. 48-Hour Backward Trajectory

The EUCLIDEN algorithm of MeteoInfo software for 48 h was used for the backward trajectory clustering of air masses. Two trajectories were classified in XN (Figure 5) and QL (Figure 6), respectively, based on total spatial variance (TSV). In XN, cluster 1 was long-distance air mass transport in the northwest, with the smallest contribution to the total trajectory (34.99%). It originated from the Badain Jaran Desert and the Tengger Desert in the west of Inner Mongolia, crossed the Ningxia, and finally reached the observation site. Cluster 2 was near-distance air mass transport in the southeast, with the shortest transport distance but the largest contribution to the total trajectory (65.01%). It originated in Henan and was greatly influenced by the southeast wind in summer. In QL, cluster 1 was long-distance air mass transport in the northwest, with the least contribution to the total trajectory (15.38%). It was affected by the central Gobi Desert, the Badain Jaran Desert, and the Tengger Desert in Inner Mongolia. Cluster 2 was local air mass transport in Shaanxi, with the shortest transport distance but the largest contribution to the total trajectory (84.62%). It was influenced by cities such as XN. In summary, the atmospheric environments of XN and QL are affected by long-distance transport in the deserts of Inner Mongolia due to the influence of wind direction. Secondly, as an outer suburb, QL is still affected by urban pollution.

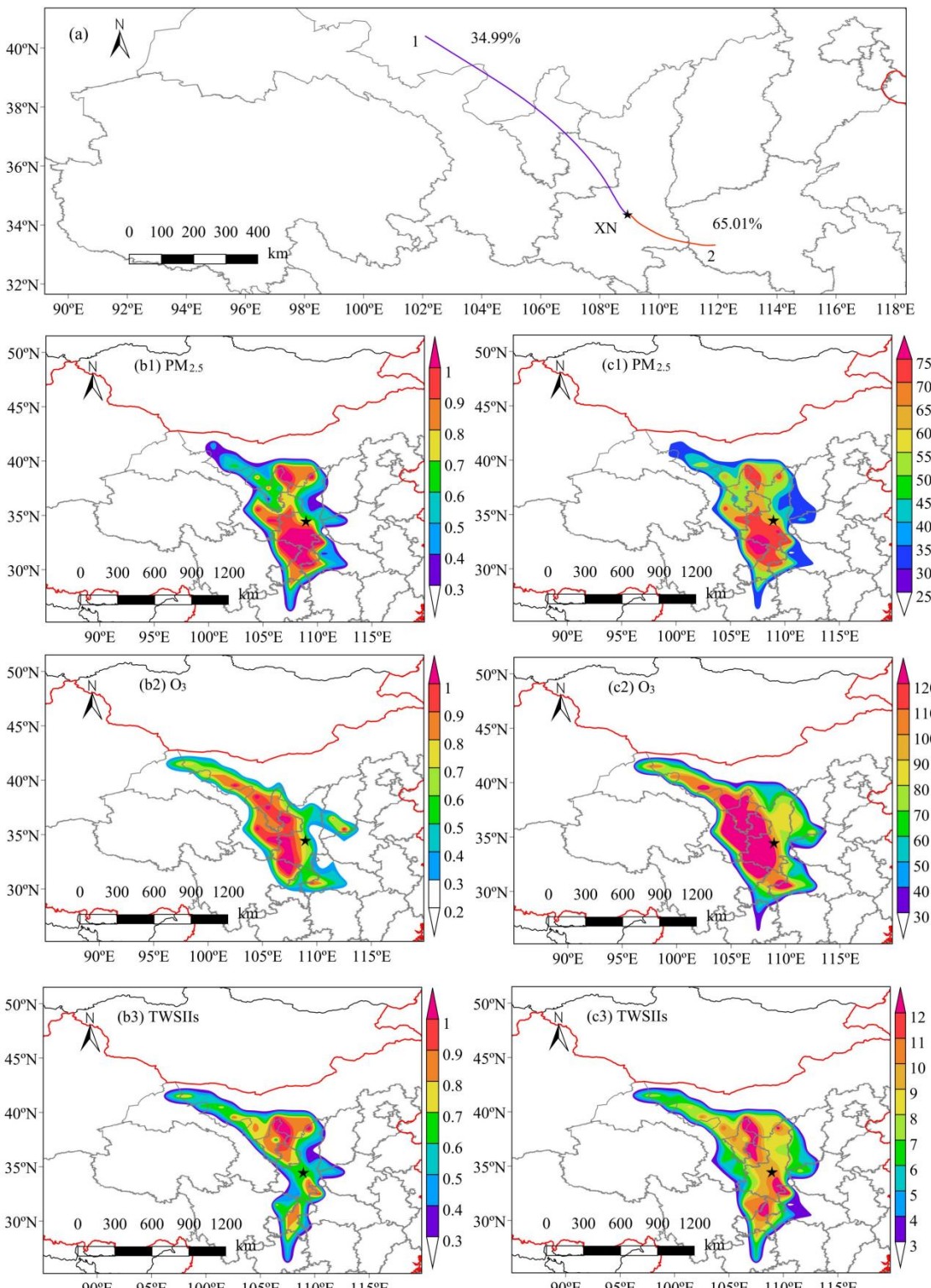

**Figure 5.** (**a**) Cluster-mean back trajectories, (**b**) WPSCF, and (**c**) WCWT of environmental parameters during the sampling periods in XN.

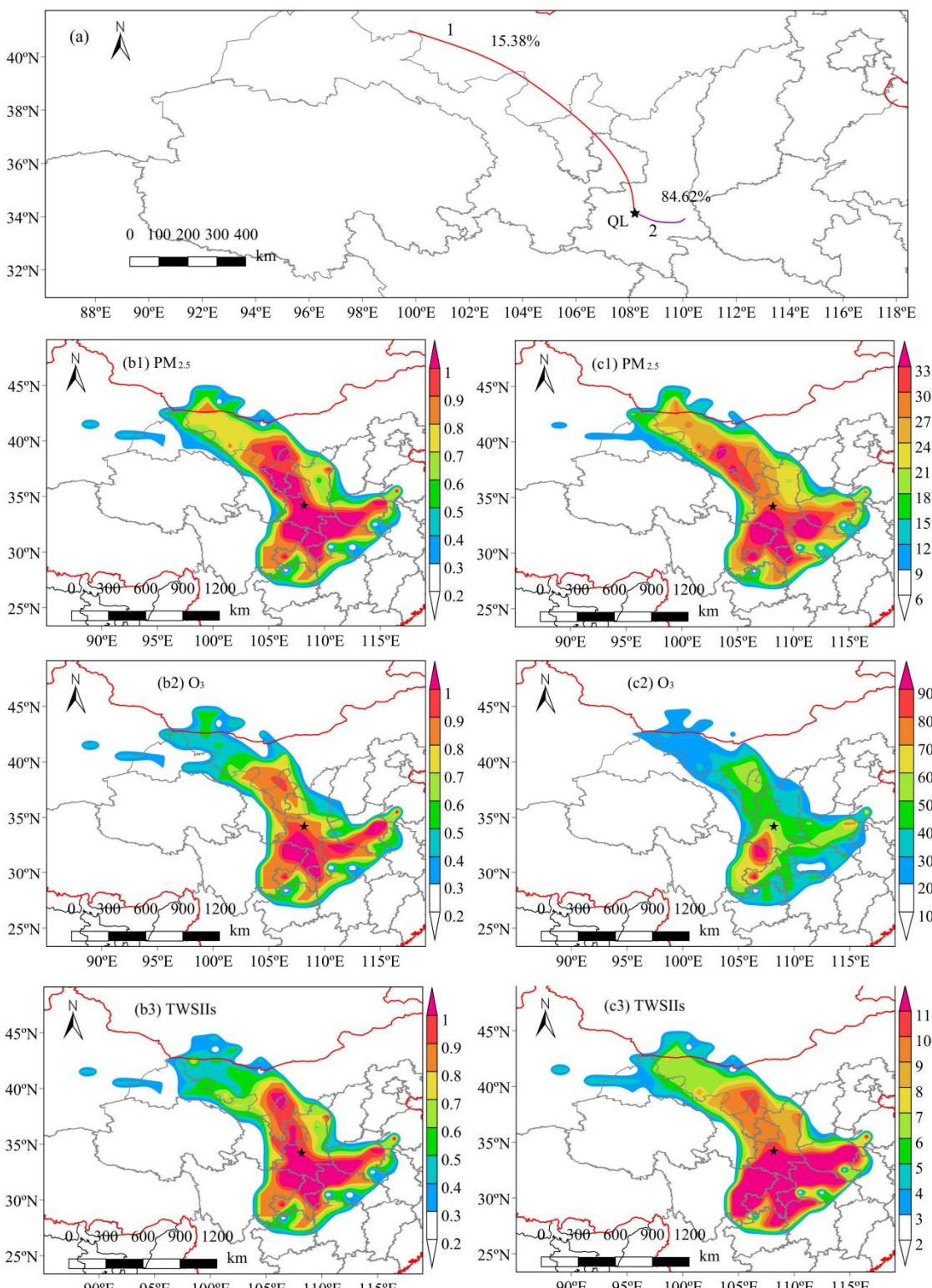

**Figure 6.** (**a**) Cluster-mean back trajectories, (**b**) WPSCF, and (**c**) WCWT of environmental parameters during the sampling periods in QL.

WPSCF and WCWT for $PM_{2.5}$, OC, EC, $O_3$, TWSIIs, $NH_4^+$, $SO_4^{2-}$, and $NO_3^-$ in XN (Figure 5) and $PM_{2.5}$, OC, EC, TWSIIs, $Na^+$, $SO_4^{2-}$, and $NH_4^+$ in QL (Figure 6) were used to evaluate the transport pathways and sources. The values of WPSCF for $PM_{2.5}$, $O_3$, and TWSIIs in XN are shown in Figure 5b; the darker the color of the column, the greater the concentration contribution of the transport air mass in the potential source area [59]. In

XN, the source areas of PM$_{2.5}$ mainly include Inner Mongolia, Gansu, Ningxia, Shaanxi, Sichuan, Chongqing, and Hubei, and the areas with WPSCF greater than 0.9 are mainly distributed in Mu Us Sandy Land; central and southern Shaanxi; and adjacent to Sichuan, Chongqing, and Hubei. The large WPSCF values of OC (Figure S6(a1)) were mainly concentrated in Inner Mongolia, Ningxia, Gansu, and Shaanxi. However, the large WPSCF values of EC (Figure S6(a2)) were mainly concentrated in Gansu, Sichuan, and Shaanxi. The source areas of O$_3$ and TWSIIs were relatively small compared with PM$_{2.5}$, and the potential source of O$_3$ was mainly distributed in Inner Mongolia, Gansu, Ningxia, Shaanxi, Sichuan, Shanxi, and Hubei, including areas of WPSCF greater than 0.9 mainly distributed in the Tengger Desert, Ningxia, Gansu, Sichuan, and Shaanxi. The source areas of heavy TWSIIs pollution were located in Mu Us Sandy Land. By comparing the WPSCF of the main ions NH$_4^+$, SO$_4^{2-}$, and NO$_3^-$ (Figure S6(a3–a5)), it is found that SO$_4^{2-}$ has the largest pollution source area and the deepest pollution degree, NH$_4^+$ has the smallest pollution source area and the smallest pollution degree, and the higher WPSCF value was only distributed around the sampling point.

As shown in Figure 6b, in QL, the main source areas of PM$_{2.5}$ were distributed in Mongolia, Inner Mongolia, Ningxia, Gansu, Shaanxi, Sichuan, Henan, Hubei, and Chongqing, and the areas with WPSCF greater than 0.9 were mainly distributed in Inner Mongolia, Ningxia, Gansu, Shaanxi, Sichuan, Chongqing, Hubei, and Henan. As shown in Figure S7(a1), the source areas of OC and PM$_{2.5}$ were generally identical. However, the high-value region of WPSCF of EC was mainly located in the southern region of the sampling site (Figure S7(a2)). The higher values of WPSCF for O$_3$ were distributed in Shaanxi, Sichuan, Chongqing, Hubei, and Henan, indicating that the NO$_X$ and VOC$_S$ emissions from heavy industry in neighboring cities affect the concentration of O$_3$ in QL (Figure 6(b2)). The higher values of WPSCF for TWSIIs were distributed in Inner Mongolia, northern Ningxia, Sichuan, Chongqing, Hubei, Henan, and Shaanxi (Figure 6(b3)). The WPSCF values of the main WSIIs, Na$^+$, NH$_4^+$, and SO$_4^{2-}$, were analyzed (Figure S7(a3–a5)), among which the Na$^+$ source had the widest pollution area and the largest pollution degree, and the high-value areas with WPSCF values greater than 0.9 were mainly distributed in Inner Mongolia, Ningxia, Gansu, Shaanxi, Sichuan, Chongqing, Hubei, and Henan. In summary, the atmospheric environments of XN and QL were affected by the Tengger Desert, Inner Mongolia, the Mu Us Desert, Gansu, and Ningxia under the action of wind power in the northwest direction, and the local pollution sources were the main sources of air pollutants in XN and QL. At the same time, XN and QL were affected by the neighboring provinces of Sichuan, Chongqing, Hubei, and Henan.

To further validate the analysis results of WPSCF, concentration weight trajectory analysis was performed using WCWT to obtain the level of contamination concentration contribution in the potential source areas of pollutants. The darker the color of the color column is, the greater the contribution of the pollutant concentration is. The values of WCWT in XN and QL are shown in Figures 5c and 6c, respectively. The color column is in µg m$^{-3}$. In XN, as shown in Figure 5(c1), WCWT values of PM$_{2.5}$ greater than 75 µg m$^{-3}$ were mainly distributed in southern Shaanxi. As shown in Figure 5(c2), the larger WCWT values of O$_3$ were distributed in Inner Mongolia, Ningxia, Gansu, Shaanxi, Sichuan, and Hubei. Compared to O$_3$ and PM$_{2.5}$, the WCWT area of TWSIIs (Figure 5(c3)) and major ions (NH$_4^+$, SO$_4^{2-}$, and NO$_3^-$) (Figure S6(b3–b5)) was smaller, and the distribution of large WCWT values was relatively scattered. In QL, as shown in Figure 6(c1), regions with WCWT values greater than 33 µg m$^{-3}$ were mainly distributed in the Tengger Desert, the south of Shaanxi, Sichuan, Chongqing, and Hubei. As shown in Figure 6(c2), the high WCWT value areas of TWSIIs were mainly distributed in the southern area of the sampling point. The results of the WCWT analysis of the major ions are shown in Figure S7(b3–b5). The WCWT of Na$^+$ had a wider range of distribution areas and contributed more to pollution concentrations in the potential source areas of pollutants than SO$_4^{2-}$ and NH$_4^+$, and the areas with high WCWT values were distributed in southern Shaanxi, Sichuan, and Henan.

Based on the clustering results of backward trajectories in XN and QL, the combined WPSCF and WCWT analysis showed that both XN and QL are affected by the desert of Inner Mongolia, the northwest regions such as Lanzhou and Ningxia, and the southern neighboring regions such as Sichuan, Chongqing, Hubei, and Henan, as well as local pollution sources. These results show that the above source areas should be focused on when formulating carbon aerosol emission policies and pollution control strategies.

*3.4. Uncertainties*

When using the PMF model for pollution source analysis, there should be no less than 20 species in principle [60]. In this study, only $PM_{2.5}$ and water-soluble ions were considered, which may lead to incomplete source analysis results. On the other hand, the PMF model is not suitable for substances with short lifespan, strong activity, or large numbers of secondary sources, which may lead to an underestimation of the relative contribution of natural and transportation sources. However, during the model operation process, we strictly adhere to the uncertainty calculation basis, which is 5% of the measured value plus the minimum detection limit. In future research, we will also consider inputting metal elements to obtain more comprehensive source apportionment results.

There are some uncertainties in the air mass trajectory model, which mainly include the following [61]: (1) numerical methods for model calculations; (2) wind field error; (3) position error and the difference between model terrain and real terrain; and (4) the chaotic nature of the atmosphere, which makes it difficult to accurately calculate trajectories. In this study, we only relied on the longitude and latitude, date, time of observation points, and meteorological data provided by GDAS, without considering the emission areas of pollution sources and estimated emissions. In recent years, the accuracy of trajectory calculation has been improved, especially in meteorological data obtained from weather forecast numerical models. Therefore, in this study, we rely on this qualitative method, which can still determine the source of pollutants. However, in the future, more refined particle dispersion models are needed to obtain quantitative analysis.

**4. Conclusions**

In this study, XN and QL were used as two typical sites in the GB to analyze the characteristics and sources of $PM_{2.5}$ and $O_3$ during the summer in the GB. The source areas and sources of carbon matter, WSIIs, and $O_3$ were discussed in detail using the PMF, WPSCF, and WCWT models combined with the back-trajectory technique. This study has contributed to our understanding of the characteristics and sources of $PM_{2.5}$ and $O_3$ pollution in the GB, providing data support for their prevention and control of $PM_{2.5}$ and $O_3$ pollution in the GB, and even global climate change and environmental sustainability. The following main conclusions were drawn.

1.  The concentration of $PM_{2.5}$ in XN (53.40 ± 17.42 μg m$^{-3}$) was higher than that in QL (27.57 ± 8.27 μg m$^{-3}$), but the concentrations of $PM_{2.5}$ in XN and QL in summer were significantly lower than in winter. TWSIIs accounted for 19.40% and 39.37% of the $PM_{2.5}$ mass in XN and QL, respectively. $NH_4^+$, $SO_4^{2-}$, and $NO_3^-$ were the most abundant in XN, accounting for 22.75%, 21.89%, and 17.00% of the TWSIIs mass, respectively; $Na^+$, $SO_4^{2-}$, and $NH_4^+$ were the most abundant in QL, accounting for 45.56%, 28.09% and 9.28% of the TWSIIs mass, respectively.
2.  The $O_3$ concentrations in summer were 102.44 ± 35.08 μg m$^{-3}$ and 47.95 ± 21.63 μg m$^{-3}$ in XN and QL, respectively, while the $O_3$ concentration and $O_x$ in summer showed a significant correlation in the GB. In the future, focus will continue to be placed on the joint prevention and control of $PM_{2.5}$ and $O_3$ pollution to control overall air pollution.
3.  The PMF model identified three sources (COB, SA, and DUSs) in XN and QL. Controlling COB may improve $PM_{2.5}$ pollution in the GB. In addition, metal and water-insoluble components can be introduced in future research to provide more detailed and accurate sources of pollution.

4.  WPSCF and WCWT with the back-trajectory analysis showed that Inner Mongolia, the interior of Shaanxi, and nearby areas to the south were the sources and source areas of carbonaceous matter in XN and QL. Therefore, it is necessary to strengthen regional joint prevention and control efforts to effectively curb $PM_{2.5}$ and $O_3$ pollution.

**Supplementary Materials:** The following supporting information can be downloaded at https://www.mdpi.com/article/10.3390/su16073074/s1: Figure S1 The wind rose diagram of XN and QL in summer. Figure S2. Correlation of OC and EC in winter and summer in XN and QL. Figure S3. The relationship between $O_3$ with $PM_{2.5}$ in QL. Figure S4. PMF source resolution map of the environmental elements in XN during the sampling periods. Figure S5. PMF source resolution map of the environmental elements in QL during the sampling periods. Figure S6. (a) The WPSCF and (b) WCWT of carbon components (OC and EC) and major WSIIs during the sampling periods in XN. Figure S7 (a) The WPSCF and (b) WCWT of carbon components (OC and EC) and major WSIIs ($Na^+$, $SO_4^{2-}$, and $NH_4^+$) during the sampling periods in QL. Table S1. Average mass concentration of 10 water-soluble ions (WSIIs) at XN and QL and the ratio to $PM_{2.5}$. Table S2. WSIIs correlation analysis in XN and QL. Table S3. Calculation results of PMF models in XN and QL during the sampling periods. Text S1. Before and after sampling, processing of filter membrane and storage of samples. Text S2. Details of Desert Research Institute (DRI) Model 2015 carbon analyzer. Text S3. The calculation method of primary organic carbon (POC) and secondary organic carbon (SOC). Text S4. Detailed pretreatment process for water-soluble ions (WSIIs). Text S5. The calculation of WSIIs acidity, alkalinity, sulfur oxidation rates (SOR) and nitrogen oxidation rates (NOR). Text S6. Post-processing of positive matrix factorization (PMF) model. Text S7. Details of potential source contribution function (PSCF) model and concentration weight trajectory (CWT) model. Text S8. Values of $W_{ij}$ in Xi'an (XN) and Qinling (QL). References [62–73] are mentioned in Supplementary Materials.

**Author Contributions:** X.L.: supervision, conceptualization, methodology, software, writing—review and editing, validation, project administration, funding acquisition; J.G.: methodology, formal analysis, writing—original draft, writing—review and editing; X.W.: methodology; Z.Y.: methodology; L.T.: writing—review and editing; F.Y.: methodology; R.Z.: methodology; W.Y.: methodology; Q.W.: data curation. All authors have read and agreed to the published version of the manuscript. All authors have read and agreed to the published version of the manuscript.

**Funding:** This work was supported by the National Natural Science Foundation of China (42177366), the State Key Laboratory of Loess and Quaternary Geology (SKLLQG2133).

**Informed Consent Statement:** This note does not contain any studies with human or animal subjects.

**Data Availability Statement:** The data presented in this study are available on request from the corresponding author.

**Conflicts of Interest:** The authors declare that they have no known competing financial interests or personal relationships that could have appeared to influence the work reported in this paper.

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
