# Peer review of "PM2.5 and O3 in an Enclosed Basin, the Guanzhong Basin of Northern China: Insights into Distributions, Appointment Sources, and Transport Pathways"

_sustainability, doi:10.3390/su16073074_

Round 1

Reviewer 1 Report

Comments and Suggestions for Authors

The authors study the air pollution sources of PM2.5 and O3 in urban and background sites of Guanzhong Basin, China, focusing on summer. The study employed PMF, WPSCF and WCWT models for qualifying and quantifying the emission sources, which is a strength of the study. The results are important and contribute to the knowledge of air pollution dynamic in one part of China. However, what is the contribution of the research to the general knowledge of air pollution?

The 3rd cluster of PMF is assigned as biomass burning, but the definition is not correct. The authors should reconsider the 3rd source. 

Moreover, the manuscript needs thorough English editing. It's difficult to understand the intention of the authors in some paragraphs.

Specific comments:

Please refer to Chinese Air Quality Guidelines or WHO AQG when stating that somewhere has air pollution.

Line 34-51: It is not clear what the authors want to say in this paragraph. 

Line 36-37: The division of PM is not correct. Some of inorganic elements are also included in WSIIs. You can divide PM into organic and inorganic, then separate by water soluble and water insoluble. 

Line 37-43: This sentence is too long.

Line 47: "...absorb the wavelengths from visible (Vis) to ultraviolet (UV) light ..."

Line 60: Do you mean "Research on O3 pollution in China was very limited."?

Line 61: what is compound pollution of PM2.5? Please use scientific terms to avoid confusion.

Line 62-63: based on which criteria did you state that "ozone pollution has been found in most parts of China". The criteria of ozone pollution should be mentioned.

Line 64-65: Please state that Xian and Qinling are in Guanzhong Basin. International readers don't know the specific area.

Line 65-69: this sentence had to 2 separate meaning. Please rewrite them.

No mention of data sources of O3.

Line 145: repeated value of OC in QL.

Line 162: respectively for which?

Line 197-200: the sentence is too long and difficult to understand.

Line 202-203: Ca2+ not Ca+

Line 216-219: XN has two values of NO3-/SO42-? Please revise this sentence.

Line 220-233: in this paragraph, the results shows that the photochemical oxidation of SO2 and NO2 only occurred in QL and not in XN. QL is clean area and XN is polluted area. Is there any relationship between location and the photochemical oxidation SO2 and NO2?

Line 284-295: Please rewrite this paragraph. It's difficult to understand what the authors want to say.

Line 305: "factor 3 was defined as a BOB due to the high loading of EC". Biomass burning source is not defined easily because of high loading of EC. You should check K+ and OC as well, as they are indicators of biomass burning. 

Line 321: Do the authors mean Trajstat?

Line 322: what TSV stands for?

Figure 2. State the names of (a)(b)(c)(d)(e)(f) graphs.

Figure 4 and 5. what b1, b2, b3, c1, c2, c3 stand for?

Reference: more than 50% of the reference are more than 5 years. Please update or remove unnecessary, old references. 

Supplement:

The equation (1) is wrong. It should be OC=OC1+OC2+OC3+OC4+OPC.

Comments on the Quality of English Language

The author should revise the manuscript to make it clear to understand. Then English editing is needed to refine the manuscript.

Author Response

Dear Editor:

     We highly appreciate your hard work and positive comments to improve our MS. We have tried our best to improve the manuscript in the current revised version according to the suggestions from the reviewer 1.

 The response to reviewer 1 and the major modifications in the manuscript have been marked in red color.

     If you have further questions or suggestions, please let me know.

                                             Yours Xiaofei Li

Response to Comments on sustainability-2864453: PM2.5 and O3 in an enclosed basin, Guanzhong Basin of Northern China: Insights into distributions, appointment sources, and transport pathways by Xiaofei Li et al.

We thank the reviewer 1 for their constructive and thoughtful comments on our manuscript.

Reviewer 1 comments:

The authors study the air pollution sources of PM2.5 and O3 in urban and background sites of Guanzhong Basin, China, focusing on summer. The study employed PMF, WPSCF and WCWT models for qualifying and quantifying the emission sources, which is a strength of the study. The results are important and contribute to the knowledge of air pollution dynamic in one part of China. However, what is the contribution of the research to the general knowledge of air pollution?

Response: To provide data support for the joint prevention and control of PM2.5 and O3 pollution in GB,furthermore, the results of this study contribute to formulating prevention and control policies and guidelines for PM2.5 and O3 in China.

The 3rd cluster of PMF is assigned as biomass burning, but the definition is not correct. The authors should reconsider the 3rd source.

Response: factor 3 was defined as BOB due to the high loading of EC and OC. This definition can be referenced in (Li, 2023)

Specific comments:

Please refer to Chinese Air Quality Guidelines or WHO AQG when stating that somewhere has air pollution.

Response: Done

Line 34-51: It is not clear what the authors want to say in this paragraph.

Response: Done

Line 36-37: The division of PM is not correct. Some of inorganic elements are also included in WSIIs. You can divide PM into organic and inorganic, then separate by water soluble and water insoluble.

Response: Aerosol of PM2.5 (particulate matter with aerodynamic diameter less than or equal to 2.5 μm), and its formation process and sources are very complex[1]. As a result, it has an important impact on the climate, the environment, and human health[2]. The main components include carbon matter, water-soluble ions (WSIIs), and metallic elements.

Line 37-43: This sentence is too long.

Response: The main components include carbon matter, water-soluble ions (WSIIs), and metallic elements. It can easily enter the human body through the respiratory system and further affect physical health. Studies indicate that PM2.5 has an impact on the respiratory system, causes cognitive impairment in the elderly, increases the risk of cancer, and causes arrhythmia; furthermore, it can lead to impaired vasoconstriction and accelerated arteriosclerosis.

Line 47: "...absorb the wavelengths from visible (Vis) to ultraviolet (UV) light ..."

Response: Done

Line 60: Do you mean "Research on O3 pollution in China was very limited."?

Response: Thank you for the suggestion. I think the description in this sentence is incorrect, so I deleted it.

Line 61: what is compound pollution of PM2.5? Please use scientific terms to avoid confusion.

Response: According to 《the Environmental Air Quality Standard》 (GB3095-2012), when the concentration of PM2.5 exceeds 75 μg m−3, it is considered PM2.5 pollution; When the average daily maximum 8h concentration of O3 is greater than 160 μg m−3, it is O3 pollution. When both PM2.5 and O3 concentrations exceed the standard, it is PM2.5 and O3 compound pollution.

Line 62-63: based on which criteria did you state that "ozone pollution has been found in most parts of China". The criteria of ozone pollution should be mentioned.

Response: According to 《the Environmental Air Quality Standard》 (GB3095-2012), When the average daily maximum 8h concentration of O3 is greater than 160 μg m−3, it is O3 pollution.

Through previous researches, O3 pollution has been found in most parts of China. The relevant references include[4]-[8]:

Line 64-65: Please state that Xian and Qinling are in Guanzhong Basin. International readers don't know the specific area.

Response: As a mega city in GB, Xi'an (XN) has severe air pollution problems. Qinling (QL) is an ecologically protected area in GB that is less polluted by human activities, therefore, a more comprehensive perspective is needed to study the characteristics of PM2.5 and O3 in GB.

Line 65-69: this sentence had to 2 separate meaning. Please rewrite them.

Response: However, research on PM2.5 and O3 in the Guanzhong Basin (GB) is relatively limited. As a mega city in GB, Xi'an (XN) has severe air pollution problems. Qinling (QL) is an ecologically protected area in GB that is less polluted by human activities, therefore, a more comprehensive perspective is needed to study the characteristics of PM2.5 and O3 in GB.

No mention of data sources of O3.

Response: In addition, the characteristics of O3 were analyzed using O3 data downloaded from National Climatic Data Center (NCDC) from 11 Aug to 11 Sep, 2021 in XN and from 14 Jul to 24 Aug, 2021 in QL. At the same time, O3 and PM2.5 data for XN and QL during winter were collected from NCDC (from 1 Jan to 31 Jan, 2021) for comparison in order to more clearly explain the characteristics of summer O3 characteristics in GB.

Line 145: repeated value of OC in QL.

Response: Done

Line 162: respectively for which?

Response: The value of OC/EC is an important index of reaction pollution sources. In summer, OC/EC values were 9.84 ± 6.95 and 10.00 ± 6.99 in XN and QL, respectively.

Line 197-200: the sentence is too long and difficult to understand.

Response: Done

Line 202-203: Ca2+ not Ca+

Response: Done

Line 216-219: XN has two values of NO3-/SO42-? Please revise this sentence.

Response: Done

Line 220-233: in this paragraph, the results shows that the photochemical oxidation of SO2 and NO2 only occurred in QL and not in XN. QL is clean area and XN is polluted area. Is there any relationship between location and the photochemical oxidation SO2 and NO2?

Response: The photochemical oxidation reaction of SO2 and NO2 is influenced by temperature, light, humidity, wind speed, and pollution source emissions. For example, when the temperature is high and the solar radiation is strong, the secondary conversion ability of SO2 and NO2 is stronger. In QL, there is more vegetation coverage, which leads to more VOCs emissions from vegetation and ultimately more SOC conversion. However, XN is densely populated with buildings, and when the wind speed is high, clean air masses will accelerate the diffusion of pollutants. Therefore, the photochemical oxidation ability of SO2 and NO2 in XN is weaker than that in QL.

Line 284-295: Please rewrite this paragraph. It's difficult to understand what the authors want to say.

Response: Done

Line 305: "factor 3 was defined as a BOB due to the high loading of EC". Biomass burning source is not defined easily because of high loading of EC. You should check K+ and OC as well, as they are indicators of biomass burning.

Response:  factor 3 was defined as BOB due to the high loading of EC and OC. This definition can be referenced in (Li, 2023)

Line 321: Do the authors mean Trajstat?

Response: The EUCLIDEN algorithm of MeteoInfo software for 48h was used for the backward trajectory clustering of air masses. Two trajectories were classified in XN (Fig.5) and QL (Fig.6), respectively, based on total spatial variance (TSV).

Line 322: what TSV stands for?

Response: TSV stands for total spatial variance.

Figure 2. State the names of (a)(b)(c)(d)(e)(f) graphs.

Response: Fig.3. The relationship between O3 and PM2.5 and meteorological factors in XN; (a), (b) the distribution of PM2.5, RH, and O3 in summer and winter, respectively; (c), (d) the distribution of O3 and TAvg in summer and winter, respectively; (e), (f) the correlation between PM2.5 and O3 in summer and winter.

Figure 4 and 5. what b1, b2, b3, c1, c2, c3 stand for?

Response: Figure 5 (a) Cluster-mean back-trajectories, (b1) WPSCF of PM2.5; (b2) WPSCF of O3; (b3) WPSCF of TWSIIs; (c1) WCWT of PM2.5; (c2) WCWT of O3; (c3) WCWT of TWSIIs during the sampling periods in XN.

Figure 6 (a) Cluster-mean back-trajectories , (b1) WPSCF of PM2.5; (b2) WPSCF of O3; (b3) WPSCF of TWSIIs; (c1) WCWT of PM2.5; (c2) WCWT of O3; (c3) WCWT of TWSIIs during the sampling periods in QL.

Reference: more than 50% of the reference are more than 5 years. Please update or remove unnecessary, old references.

Response: Done

Supplement:

The equation (1) is wrong. It should be OC=OC1+OC2+OC3+OC4+OPC.

Response: Done

References

[1] Alves, C.; Evtyugina, M.; Vicente, E.; Vicente, A.; Rienda, I. C.; de la Campa, A. S.; Tomé, M.; Duarte, I., PM2.5 chemical composition and health risks by inhalation near a chemical complex. Journal of Environmental Sciences 2023, 124, 860-874.

[2] Jerrett, M., The death toll from air-pollution sources. Nature 2015, 525, (7569), 330-331.

[3] Li X, Guo J, Yu F, et al. Concentrations, sources, fluxes, and absorption properties of carbonaceous matter in a central Tibetan Plateau river basin[J]. Environmental Research, 2023, 216: 114680.

[4] Zhao, S.; Yin, D.; Yu, Y.; Kang, S.; Qin, D.; Dong, L., PM2.5 and O3 pollution during 2015–2019 over 367 Chinese cities: Spatiotemporal variations, meteorological and topographical impacts. Environmental Pollution 2020, 264, 114694.

[5] Liu, J.; Wang, L.; Li, M.; Liao, Z.; Sun, Y.; Song, T.; Gao, W.; Wang, Y.; Li, Y.; Ji, D., Quantifying the impact of synoptic circulation patterns on ozone variability in northern China from April to October 2013–2017. Atmospheric Chemistry and Physics 2019, 19, (23), 14477-14492.

[6] Wang, L.; Zhao, B.; Zhang, Y.; Hu, H., Correlation between surface PM2.5 and O3 in eastern China during 2015–2019: Spatiotemporal variations and meteorological impacts. Atmospheric Environment 2023, 294, 119520.

[7] Zhao, H.; Zheng, Y.; Li, C., Spatiotemporal distribution of PM2.5 and O3 and their interaction during the summer and winter seasons in Beijing, China. Sustainability 2018, 10, (12), 4519.

[8] Xu, T.; Zhang, C.; Liu, C.; Hu, Q., Variability of PM2.5 and O3 concentrations and their driving forces over Chinese megacities during 2018-2020. Journal of Environmental Sciences 2023, 124, 1-10.

Reviewer 2 Report

Comments and Suggestions for Authors

Manuscript Number: sustainability-2864453
Title: PM2.5 and O3 in an enclosed basin, Guanzhong Basin of Northern China: Insights into distributions, appointment sources, and transport pathways

Comments to the authors:

The authors tackle an interesting research topic centered around air quality assessment, particularly in identifying distributions, sources, and transport pathways for PM2.5 and O3. The manuscript aligns well with the journal's scope, demonstrating commendable writing, presentation, and organization. To further elevate the manuscript quality, it would be beneficial to underscore its novelty in the research field. Explicitly highlighting the contributions this work makes to the existing literature will provide valuable insights for future studies in the area. Therefore, with some minor revisions, I am confident that this manuscript is ready for publication.

When submitting the revised manuscript, please take note of the following: Conclusions must be reformulated. It is recommended to succinctly present the main findings, emphasizing the research's novelty and relevance for the field.

Author Response

Dear Editor:

     We highly appreciate your hard work and positive comments to improve our MS. We have tried our best to improve the manuscript in the current revised version according to the suggestions from the reviewer 2.

 The response to reviewer 2 and the major modifications in the manuscript have been marked in red color.

     If you have further questions or suggestions, please let me know.

                                             Yours Xiaofei Li

Response to Comments on sustainability-2864453: PM2.5 and O3 in an enclosed basin, Guanzhong Basin of Northern China: Insights into distributions, appointment sources, and transport pathways by Xiaofei Li et al.

We thank the reviewer 2 for their constructive and thoughtful comments on our manuscript.

Reviewer 2 comments:

The authors tackle an interesting research topic centered around air quality assessment, particularly in identifying distributions, sources, and transport pathways for PM2.5 and O3. The manuscript aligns well with the journal's scope, demonstrating commendable writing, presentation, and organization. To further elevate the manuscript quality, it would be beneficial to underscore its novelty in the research field. Explicitly highlighting the contributions this work makes to the existing literature will provide valuable insights for future studies in the area. Therefore, with some minor revisions, I am confident that this manuscript is ready for publication.

When submitting the revised manuscript, please take note of the following: Conclusions must be reformulated. It is recommended to succinctly present the main findings, emphasizing the research's novelty and relevance for the field.

Response: In this study, XN and QL were used as two typical sites in the GB to analyze the characteristics and sources of PM2.5 and O3 during the summer in GB. The source areas and source of carbon matter, WSIIs, and O3 were discussed in detail using the PMF, WPSCF, and WCWT models combined with the back-trajectory technique. This study has contributed to our understanding of the characteristics and sources of PM2.5 and O3 pollution in GB, providing data support for their prevention and control of PM2.5 and O3 pollution in GB. The following main conclusions were drawn.

(1) The concentration of PM2.5 was higher in XN (53.40 ± 17.42 μg m−3) in summer compared with QL (27.57 ± 8.27 μg m−3), but it was lower in winter. TWSIIs accounted for 19.40% and 39.37% of the PM2.5 mass in XN and QL, respectively. NH4+, SO42−, and NO3 were the most abundant in XN, accounting for 22.75%, 21.89%, and 17.00% of the TWSIIs mass, respectively; Na+, SO42−, and NH4+ were the most abundant in QL, accounting for 45.56%, 28.09% and 9.28% of the TWSIIs mass, respectively.

(2) The O3 concentrations in summer were 102.44 ± 35.08 μg m−3 and 47.95 ± 21.63 μg m−3 in XN and QL, respectively, while the O3 concentration and OX in summer showed a significant correlation in GB. In the future, focus will continue to be placed on the joint prevention and control of PM2.5 and O3 pollution to control overall air pollution.

(3) The PMF model identified three sources (BOB, SA, and DUSs) in XN and QL. Controlling BOB may improve PM2.5 pollution in GB. In addition, metal and water-insoluble components can be introduced in future research to provide more detailed and accurate sources of pollution.

(4) WPSCF and WCWT with the back-trajectory analysis showed that Inner Mongolia, the interior of Shaanxi, and nearby areas to the south were the source and source areas of carbonaceous matter in XN and QL. Therefore, it is necessary to strengthen regional joint prevention and control efforts to effectively curb PM2.5 and O3 pollution.

Reviewer 3 Report

Comments and Suggestions for Authors

1. Line 16: PM2.5 is one kind of aerosol samples, and I suggest the authors to describe more accurately as follows: Aerosol of PM2.5 (particulate matter with aerodynamic diameter less than 2.5 μm).

2. Please add a topographic map of the research area and list the sampling site locations in section 2.1, which will be more beneficial for readers to understand the study area.

3. Line 131 and line 320: Pay attention to the format.

4. Lines 266-274: The relationship between O3 and PM2.5 is complex, and I suggest the authors consult literature to add discussion and explanation in this section.

5. The Guanzhong Basin, especially Xi'an City is the basin terrain, which adjacent to the Loess Plateau to the north and the Qinling Mountains to the south. Does terrain have an impact on the diffusion of O3?

6. Please list the limitations of the manuscript in the conclusion section.

Author Response

Dear Editor:

     We highly appreciate your hard work and positive comments to improve our MS. We have tried our best to improve the manuscript in the current revised version according to the suggestions from the reviewer 3.

 The response to reviewer 3 and the major modifications in the manuscript have been marked in red color.

     If you have further questions or suggestions, please let me know.

                                             Yours Xiaofei Li

Response to Comments on sustainability-2864453: PM2.5 and O3 in an enclosed basin, Guanzhong Basin of Northern China: Insights into distributions, appointment sources, and transport pathways by Xiaofei Li et al.

We thank the reviewer 3 for their constructive and thoughtful comments on our manuscript.

Reviewer 3 comments:

Line 16: PM2.5 is one kind of aerosol samples, and I suggest the authors to describe more accurately as follows: Aerosol of PM2.5 (particulate matter with aerodynamic diameter less than 2.5 μm).

Response: Aerosol of PM2.5 (particulate matter with aerodynamic diameter less than or equal to 2.5 μm).

Please add a topographic map of the research area and list the sampling site locations in section 2.1, which will be more beneficial for readers to understand the study area.

Response:

Fig.1 Location map of sampling points

Line 131 and line 320: Pay attention to the format.

Response: Done

Lines 266-274: The relationship between O3 and PM2.5 is complex, and I suggest the authors consult literature to add discussion and explanation in this section.

Response: Sulfate, nitrate, and carbonaceous aerosols in PM2.5 can directly scatter or absorb solar radiation, indirectly changing the optical properties and life cycle of clouds and affecting the intensity of ultraviolet radiation intensity, and thus affecting the photolysis rate and generation of O3[53]. A high concentration of O3 means that there is strong photochemical reactivity in the atmosphere and promotes the formation of SOC. For example, when the photochemical reaction in the atmosphere is strong, SO2 will be photochemically oxidized to H2SO4, and when the O3 concentration is high, it will promote the conversion of NO2 to HNO3. In this study, PM2.5 and O3 were positively correlated in summer, and their concentrations synchronously increased or decreased overall. The main reason for this is that the temperature is high and the radiation is strong in summer, and the atmospheric particles enhance the downward radiation flux of the sun through the scattering of aerosol particles, which is conducive to the generation of O3. In addition, a high O3 concentration, strong light, and enhanced atmospheric oxidation further promote the generation of SOC[54].

[53] Zhu, J.; Chen, L.; Liao, H.; Dang, R., Correlations between PM2.5 and ozone over China and associated underlying reasons. Atmosphere 2019, 10, (7), 352.

[54] Zhang, J.; Li, Y.; Liu, C.; Wu, B.; Shi, K., A study of cross-correlations between PM2.5 and O3 based on Copula and Multifractal methods. Physica A: Statistical Mechanics and Its Applications 2022, 589, 126651.

The Guanzhong Basin, especially Xi'an City is the basin terrain, which adjacent to the Loess Plateau to the north and the Qinling Mountains to the south. Does terrain have an impact on the diffusion of O3?

Response: XN, which adjacent to the Loess Plateau to the north and the Qinling Mountains to the south, its unique geographical location condition is favorable to the formation and accumulation of severe air pollutants.

Please list the limitations of the manuscript in the conclusion sectio

Response: In this study, XN and QL were used as two typical sites in the GB to analyze the characteristics and sources of PM2.5 and O3 during the summer in GB. The source areas and source of carbon matter, WSIIs, and O3 were discussed in detail using the PMF, WPSCF, and WCWT models combined with the back-trajectory technique. This study has contributed to our understanding of the characteristics and sources of PM2.5 and O3 pollution in GB, providing data support for their prevention and control of PM2.5 and O3 pollution in GB. The following main conclusions were drawn.

(1) The concentration of PM2.5 was higher in XN (53.40 ± 17.42 μg m−3) in summer compared with QL (27.57 ± 8.27 μg m−3), but it was lower in winter. TWSIIs accounted for 19.40% and 39.37% of the PM2.5 mass in XN and QL, respectively. NH4+, SO42−, and NO3 were the most abundant in XN, accounting for 22.75%, 21.89%, and 17.00% of the TWSIIs mass, respectively; Na+, SO42−, and NH4+ were the most abundant in QL, accounting for 45.56%, 28.09% and 9.28% of the TWSIIs mass, respectively.

(2) The O3 concentrations in summer were 102.44 ± 35.08 μg m−3 and 47.95 ± 21.63 μg m−3 in XN and QL, respectively, while the O3 concentration and OX in summer showed a significant correlation in GB. In the future, focus will continue to be placed on the joint prevention and control of PM2.5 and O3 pollution to control overall air pollution.

(3) The PMF model identified three sources (BOB, SA, and DUSs) in XN and QL. Controlling BOB may improve PM2.5 pollution in GB. In addition, metal and water-insoluble components can be introduced in future research to provide more detailed and accurate sources of pollution.

(4) WPSCF and WCWT with the back-trajectory analysis showed that Inner Mongolia, the interior of Shaanxi, and nearby areas to the south were the source and source areas of carbonaceous matter in XN and QL. Therefore, it is necessary to strengthen regional joint prevention and control efforts to effectively curb PM2.5 and O3 pollution.

References

[53] Zhu, J.; Chen, L.; Liao, H.; Dang, R., Correlations between PM2.5 and ozone over China and associated underlying reasons. Atmosphere 2019, 10, (7), 352.

[54] Zhang, J.; Li, Y.; Liu, C.; Wu, B.; Shi, K., A study of cross-correlations between PM2.5 and O3 based on Copula and Multifractal methods. Physica A: Statistical Mechanics and Its Applications 2022, 589, 126651.

Reviewer 4 Report

Comments and Suggestions for Authors

The study conducted by Li et al. aimed to investigate the characteristics and sources of aerosol samples, specifically focusing on PM2.5 and ozone (O3) data collected in Xi'an (XN) and Qinling (QL) within the Guanzhong Basin (GB), China. Their findings revealed notable disparities between XN and QL, with higher PM2.5 concentrations, organic carbon (OC), and element carbon (EC) observed in XN during the summer season. Conversely, total water-soluble ions (TWSIIs) exhibited a greater impact on PM2.5 levels in QL. Utilizing Positive Matrix Factorization (PMF) analysis, the authors identified biomass burning source (BOB), secondary aerosol (SA), and dust source (DUSs) as primary contributors to aerosol pollution in both regions. While acknowledging the authors' efforts, several critical areas requiring improvement have been identified.   

-  it is recommended to include a paragraph discussing the toxicity of particulate matter and its health impacts in the introduction section to provide a comprehensive overview of the study's significance.

 - The inclusion of a map illustrating the geographical locations of XN and QL would facilitate readers' comprehension of the study area.

  - Detailed descriptions of the location characteristics, including altitude and meteorological data, should be incorporated to enhance contextual understanding.

.- The manuscript lacks adequate quality control measures and meteorological data, which are essential for ensuring the reliability and validity of the findings. 

-- The extensive use of Supplementary Information (SI) files without sufficient summarization in the main manuscript detracts from the accessibility and clarity of the study. 

 -- Regarding the choice of methodology, consideration should be given to using Principal Component Analysis (PCA) instead of PMF for source apportionment, especially given the relatively low number of samples. 

-- Detailed protocols for quality control and data collection, including information on blank filters, should be provided in the SI. 

-- PMF quality modeling data, such as uncertainty analysis, should be included to ensure the robustness of the results.  

-- The absence of data on metals and water-insoluble components is noted, which may significantly impact the PMF outcomes. Such limitations should be clearly mentioned. 

-- Regarding the location-specific characteristics, it is puzzling that the PMF results did not reflect the proximity of the XN sampling site to traffic sources, warranting further explanation.  

-- Statistical significance should be demonstrated using p-values for comparison analyses. 

-- Revisions to the naming conventions of figures are recommended for clarity.  

-- The inclusion of wind rose diagrams for both locations would provide valuable insights into the prevailing wind patterns and their influence on aerosol dispersion. 

-- It is imperative that the discussion of the results delves deeper into their implications and significance within the broader scientific context. Rather than solely focusing on numerical comparisons, the discussion should strive to provide a comprehensive interpretation of the findings, drawing upon previous research and theoretical frameworks to elucidate underlying mechanisms and trends. By contextualizing the results within existing literature, the authors can offer more nuanced insights into the dynamics of aerosol pollution in the Guanzhong Basin and contribute to advancing our understanding of environmental processes. 

-- The inclusion of comprehensive details regarding all data inputs utilized in the Positive Matrix Factorization (PMF) analysis is crucial for ensuring transparency and reproducibility in scientific research. Therefore, it is recommended that the manuscript not only provides a thorough description of the raw data but also summarizes this information in a table format. This table should include key parameters such as the mean values and standard deviations of the variables used as input to the PMF model. By presenting this information in a concise and structured manner, readers can easily grasp the characteristics of the dataset and assess the reliability of the PMF results.

Comments on the Quality of English Language

English grammatical and type issues are detected. 

Author Response

Dear Editor:

     We highly appreciate your hard work and positive comments to improve our MS. We have tried our best to improve the manuscript in the current revised version according to the suggestions from the reviewer 4.

 The response to reviewer 4 and the major modifications in the manuscript have been marked in red color.

     If you have further questions or suggestions, please let me know.

                                             Yours Xiaofei Li

Response to Comments on sustainability-2864453: PM2.5 and O3 in an enclosed basin, Guanzhong Basin of Northern China: Insights into distributions, appointment sources, and transport pathways by Xiaofei Li et al.

We thank the reviewer 4 for their constructive and thoughtful comments on our manuscript.

Reviewer 4 comments:

The study conducted by Li et al. aimed to investigate the characteristics and sources of aerosol samples, specifically focusing on PM2.5 and ozone (O3) data collected in Xi'an (XN) and Qinling (QL) within the Guanzhong Basin (GB), China. Their findings revealed notable disparities between XN and QL, with higher PM2.5 concentrations, organic carbon (OC), and element carbon (EC) observed in XN during the summer season. Conversely, total water-soluble ions (TWSIIs) exhibited a greater impact on PM2.5 levels in QL. Utilizing Positive Matrix Factorization (PMF) analysis, the authors identified biomass burning source (BOB), secondary aerosol (SA), and dust source (DUSs) as primary contributors to aerosol pollution in both regions. While acknowledging the authors' efforts, several critical areas requiring improvement have been identified.

Specific comments:

it is recommended to include a paragraph discussing the toxicity of particulate matter and its health impacts in the introduction section to provide a comprehensive overview of the study's significance.

Response: It can easily enter the human body through the respiratory system and further affect physical health. Studies indicate that PM2.5 has an impact on the respiratory system, causes cognitive impairment in the elderly, increases the risk of cancer, and causes arrhythmia; furthermore, it can lead to impaired vasoconstriction and accelerated arteriosclerosis[3].

The inclusion of a map illustrating the geographical locations of XN and QL would facilitate readers' comprehension of the study area.

Response:

Fig.1 Location map of sampling points

Detailed descriptions of the location characteristics, including altitude and meteorological data, should be incorporated to enhance contextual understanding.

Response: XN is located in the GB city cluster in Northwest China, and its unique geographical location conditions are favorable for the formation and accumulation of severe air pollutants (Fig.1). XN has a warm temperate semi humid continental monsoon climate, with distinct cold, warm, dry, and wet seasons. Summer is hot and rainy, with prevailing southeast winds; winter is dry and cold, with prevailing northwest winds. Until 2020, it is a mega city that covers 10752 Km2 with a population of around 130 million. The city is located at an altitude of around 400 meters. The sampling site was located on the roof of a building in XN (108.97°E, 34.37°N; ~30 above ground level). The building is adjacent to the highway and surrounded by teaching areas and student living areas. There are no large industrial emission sources around the sampling site.

QL is located in the south of the GB; it is an important ecological protection area in the region and has important ecological functions (Fig.1). The city is located at an average altitude of 2400 meters. The sampling site in this city was located in the suburbs far from the city (108.22°E, 34.16°N). The surrounding vegetation is abundant, and human pollution is relatively low.

The manuscript lacks adequate quality control measures and meteorological data, which are essential for ensuring the reliability and validity of the findings.

Response: Before sampling, the quartz microfiber filters (Φ90 mm) pre-baked at 450℃ for 4 h to remove residual organics and other impurities; during sampling, wear mask gloves and used clean tweezers to clip the filter membrane; after collection, the samples were individually sealed and preserved in darkness at -20℃ for further analysis. Before and after sampling, filters was weighted using an analytical balance after balancing for 48h in a drying dish. At the beginning and end of sampling, collect one blank sample as a control.

To ensure the reliability and accuracy of the findings, we considered meteorological parameters, including temperature, relative humidity, wind direction, and wind speed in this study.

The extensive use of Supplementary Information (SI) files without sufficient summarization in the main manuscript detracts from the accessibility and clarity of the study.

Response: Done

Regarding the choice of methodology, consideration should be given to using Principal Component Analysis (PCA) instead of PMF for source apportionment, especially given the relatively low number of samples.

Response: PMF has been widely used in the analysis of aerosol pollution sources, even in small sample sizes. The relevant references include the following content [1] and [2]:

Detailed protocols for quality control and data collection, including information on blank filters, should be provided in the S1.

Response: PM2.5 was continuously collected by the medium flow sampler (HC-1010, Qingdao, China) at a flow rate of 100 L min1 from 08:00 a.m. to 07:00 a.m. the next day (Beijing time). Before sampling, the quartz microfiber filters (Φ90 mm) pre-baked at 450℃ for 4 h to remove residual organics and other impurities; during sampling, wear mask gloves and used clean tweezers to clip the filter membrane; after collection, the samples were individually sealed and preserved in darkness at -20℃ for further analysis. Before and after sampling, filters was weighted using an analytical balance after balancing for 48h in a drying dish. At the beginning and end of sampling, collect one blank sample as a control.

PMF quality modeling data, such as uncertainty analysis, should be included to ensure the robustness of the results. 

Response: Table S3 Calculation results of PMF models in XN and QL during the sampling periods.

QL

Factor Profiles (conc. of species)

Factor Profiles (% of species sum)

Species

Factor 1

Factor 2

Factor 3

Factor 1

Factor 2

Factor 3

F

0.049933

0.04073

0.055551

34.15062853

27.85642962

37.99294185

Cl

0.18044

0.073885

0.46224

25.18124664

10.31099761

64.50775575

SO42

0.47965

0.46341

2.5685

13.65917142

13.1966989

73.14412967

NO3

0

0.30234

2.6666

0

10.18343247

89.81656753

Na+

0.10286

0.058562

0.034765

52.42956975

29.850092

17.72033825

K+

0.084382

0.050448

0.14979

29.64724896

17.72468555

52.62806549

Mg2+

0.065053

0.010207

0.022148

66.78404238

10.47860545

22.73735217

Ca2+

0.90573

0.058613

0.12142

83.41875713

5.398323575

11.18291929

PM2.5

24.663

19.101

0

56.35453798

43.64546202

0

OC

0.61336

3.0126

0.35057

15.42450327

75.75951898

8.815977749

EC

0.0095946

0.52202

0.077896

1.574148177

85.64576235

12.78008947

QL

Factor Profiles (conc. of species)

Factor Profiles (% of species sum)

Species

Factor 1

Factor 2

Factor 3

Factor 1

Factor 2

Factor 3

F

0.018146

0.021822

0.014076

33.5763452

40.37821035

26.04544445

Cl

0.11122

0.054291

0.052556

51.00267349

24.89647677

24.10084974

SO42

0.66223

2.5046

0.51302

17.99611397

68.0625569

13.94132913

NO3

0.18989

0.4376

0

30.26183684

69.73816316

0

Na+

2.5925

0.84314

2.2235

45.81084758

14.8987302

39.29042222

NH4+

0

1.1092

0

0

100

0

K+

0.11266

0.056288

0.11295

39.96480997

19.96750598

40.06768406

Mg2+

0.044789

0.0012963

0.013331

75.38167136

2.181724544

22.4366041

Ca2+

0.35419

0.0092232

0.14671

69.43224696

1.808033824

28.75971922

PM2.5

13.005

3.9393

16.803

38.53641625

11.67293383

49.79064992

OC

1.814

0.50916

2.9339

34.50597863

9.685261344

55.80876003

EC

0

0

0.72777

0

0

100

The absence of data on metals and water-insoluble components is noted, which may significantly impact the PMF outcomes. Such limitations should be clearly mentioned.

Response: When using the PMF model for pollution source analysis, there should be no less than 20 species in principle. In this study, only PM2.5 and water-soluble ions were considered, which may lead to incomplete source analysis results. On the other hand, the PMF model is not suitable for substances with short lifespan, strong activity, or a large number of secondary sources, which may lead to an underestimation of the relative contribution of natural and transportation sources. In future research, we will also consider inputting metal elements to obtain more comprehensive source apportionment results.

Regarding the location-specific characteristics, it is puzzling that the PMF results did not reflect the proximity of the XN sampling site to traffic sources, warranting further explanation.

Response: When using the PMF model for pollution source analysis, there should be no less than 20 species in principle. In this study, only PM2.5 and water-soluble ions were considered, which may lead to incomplete source analysis results. On the other hand, the PMF model is not suitable for substances with short lifespan, strong activity, or a large number of secondary sources, which may lead to an underestimation of the relative contribution of natural and transportation sources. This may be the reason why the PMF results did not reflect the proximity of the XN sampling location to the traffic source.

Statistical significance should be demonstrated using p-values for comparison analyses.

Response: Done

Revisions to the naming conventions of figures are recommended for clarity.

Response: Thank you for your suggestion. I have made modifications to all the naming conventions of figures. For example:

Fig.2. WSIIs concentrations and correlations. (a) concentration of ions; (b), (c), (d), (e) and (f) correlation of the main ions.

Fig.3. The relationship between O3 and PM2.5 and meteorological factors in XN; (a), (b) the distribution of PM2.5, RH, and O3 in summer and winter, respectively; (c), (d) the distribution of O3 and TAvg in summer and winter, respectively; (e), (f) the correlation between PM2.5 and O3 in summer and winter.

The inclusion of wind rose diagrams for both locations would provide valuable insights into the prevailing wind patterns and their influence on aerosol dispersion.

Response: The wind rose diagram is shown in Fig.S1.

Fig.S1 The wind rose diagram of XN and QL in summer

It is imperative that the discussion of the results delves deeper into their implications and significance within the broader scientific context. Rather than solely focusing on numerical comparisons, the discussion should strive to provide a comprehensive interpretation of the findings, drawing upon previous research and theoretical frameworks to elucidate underlying mechanisms and trends. By contextualizing the results within existing literature, the authors can offer more nuanced insights into the dynamics of aerosol pollution in the Guanzhong Basin and contribute to advancing our understanding of environmental processes.

Response: In this study, XN and QL were used as two typical sites in the GB to analyze the characteristics and sources of PM2.5 and O3 during the summer in GB. The source areas and source of carbon matter, WSIIs, and O3 were discussed in detail using the PMF, WPSCF, and WCWT models combined with the back-trajectory technique. This study has contributed to our understanding of the characteristics and sources of PM2.5 and O3 pollution in GB, providing data support for their prevention and control of PM2.5 and O3 pollution in GB. The following main conclusions were drawn.

(1) The concentration of PM2.5 was higher in XN (53.40 ± 17.42 μg m−3) in summer compared with QL (27.57 ± 8.27 μg m−3), but it was lower in winter. TWSIIs accounted for 19.40% and 39.37% of the PM2.5 mass in XN and QL, respectively. NH4+, SO42−, and NO3 were the most abundant in XN, accounting for 22.75%, 21.89%, and 17.00% of the TWSIIs mass, respectively; Na+, SO42−, and NH4+ were the most abundant in QL, accounting for 45.56%, 28.09% and 9.28% of the TWSIIs mass, respectively.

(2) The O3 concentrations in summer were 102.44 ± 35.08 μg m−3 and 47.95 ± 21.63 μg m−3 in XN and QL, respectively, while the O3 concentration and OX in summer showed a significant correlation in GB. In the future, focus will continue to be placed on the joint prevention and control of PM2.5 and O3 pollution to control overall air pollution.

(3) The PMF model identified three sources (BOB, SA, and DUSs) in XN and QL. Controlling BOB may improve PM2.5 pollution in GB. In addition, metal and water-insoluble components can be introduced in future research to provide more detailed and accurate sources of pollution.

(4) WPSCF and WCWT with the back-trajectory analysis showed that Inner Mongolia, the interior of Shaanxi, and nearby areas to the south were the source and source areas of carbonaceous matter in XN and QL. Therefore, it is necessary to strengthen regional joint prevention and control efforts to effectively curb PM2.5 and O3 pollution.

The inclusion of comprehensive details regarding all data inputs utilized in the Positive Matrix Factorization (PMF) analysis is crucial for ensuring transparency and reproducibility in scientific research. Therefore, it is recommended that the manuscript not only provides a thorough description of the raw data but also summarizes this information in a table format. This table should include key parameters such as the mean values and standard deviations of the variables used as input to the PMF model. By presenting this information in a concise and structured manner, readers can easily grasp the characteristics of the dataset and assess the reliability of the PMF results.

Response:

Table S3 Calculation results of PMF models in XN and QL during the sampling periods.

QL

Factor Profiles (conc. of species)

Factor Profiles (% of species sum)

Species

Factor 1

Factor 2

Factor 3

Factor 1

Factor 2

Factor 3

F

0.049933

0.04073

0.055551

34.15062853

27.85642962

37.99294185

Cl

0.18044

0.073885

0.46224

25.18124664

10.31099761

64.50775575

SO42

0.47965

0.46341

2.5685

13.65917142

13.1966989

73.14412967

NO3

0

0.30234

2.6666

0

10.18343247

89.81656753

Na+

0.10286

0.058562

0.034765

52.42956975

29.850092

17.72033825

K+

0.084382

0.050448

0.14979

29.64724896

17.72468555

52.62806549

Mg2+

0.065053

0.010207

0.022148

66.78404238

10.47860545

22.73735217

Ca2+

0.90573

0.058613

0.12142

83.41875713

5.398323575

11.18291929

PM2.5

24.663

19.101

0

56.35453798

43.64546202

0

OC

0.61336

3.0126

0.35057

15.42450327

75.75951898

8.815977749

EC

0.0095946

0.52202

0.077896

1.574148177

85.64576235

12.78008947

QL

Factor Profiles (conc. of species)

Factor Profiles (% of species sum)

Species

Factor 1

Factor 2

Factor 3

Factor 1

Factor 2

Factor 3

F

0.018146

0.021822

0.014076

33.5763452

40.37821035

26.04544445

Cl

0.11122

0.054291

0.052556

51.00267349

24.89647677

24.10084974

SO42

0.66223

2.5046

0.51302

17.99611397

68.0625569

13.94132913

NO3

0.18989

0.4376

0

30.26183684

69.73816316

0

Na+

2.5925

0.84314

2.2235

45.81084758

14.8987302

39.29042222

NH4+

0

1.1092

0

0

100

0

K+

0.11266

0.056288

0.11295

39.96480997

19.96750598

40.06768406

Mg2+

0.044789

0.0012963

0.013331

75.38167136

2.181724544

22.4366041

Ca2+

0.35419

0.0092232

0.14671

69.43224696

1.808033824

28.75971922

PM2.5

13.005

3.9393

16.803

38.53641625

11.67293383

49.79064992

OC

1.814

0.50916

2.9339

34.50597863

9.685261344

55.80876003

EC

0

0

0.72777

0

0

100

References

[1] Li X, Guo J, Yu F, et al. Concentrations, sources, fluxes, and absorption properties of carbonaceous matter in a central Tibetan Plateau river basin[J]. Environmental Research, 2023, 216: 114680.

[2] Huang X, Yu H, Tong L, et al. Real-time non-refractory PM1 chemical composition, size distribution and source apportionment at a coastal industrial park in the Yangtze River Delta region, China[J]. Science of The Total Environment, 2021, 763: 142968.

[3] Hua Jing, H. J.; Yin Yong, Y. Y.; Peng Li, P. L.; Du Li, D. L.; Geng Fuhai, G. F.; Zhu Liping, Z. L., Acute effects of black carbon and PM2.5 on children asthma admissions: a time-series study in a Chinese city. 2014, 481: 433-438.

Round 2

Reviewer 1 Report

Comments and Suggestions for Authors

Thank you for your effort to revise the manuscript. 

The manuscript revised according to the comments of reviewer. The manuscript has been improved much better. 

But biomass burning factor is a big problem in my recognition. I have checked the reference you mentioned. In Li et al (2023), they mentioned that "A factor with the relatively higher load of K+, Na+, and Cl− is defined as BOB." So, your argument of high loading of EC and OC cannot stand. 

From my knowledge, you can check the OC/EC (60.3) and Char-EC/Soot-EC (>10), to see if they are from BOB or not. More information is from the following paper: Characteristics and sources of the fine carbonaceous aerosols in Haikou, China - ScienceDirect

Your results showed that BOB occupied for 26-38% of PM2.5, which is a very large source. The sampling period was in summer, so it's very low chance there's a large source of BOB. There should be more evidence to support your classification such as hotspot during sampling time. 

So, my advice is to check OC/EC and Char-EC/Soot-EC in your data, and hotspot during your sampling period to confirm factor 3. Otherwise, you can name factor 3 as Others.

Some minor comments:

Line 140: 2.4.248. h back trajectory ->2.4.2. 48-hour backward trajectory

Line 344 3.3.248. h back trajectory

Line 440-441: do the authors mean the PM2.5 in XN was lower than QL in winter?

Author Response

Dear Editor:

  We highly appreciate your hard work and positive comments to improve our MS. We have tried our best to improve the manuscript in the current revised version according to the suggestions from the reviewer 1.

  The response to reviewer 1 and the major modifications in the manuscript have been marked in red color.

  If you have further questions or suggestions, please let me know.

                                             Yours Xiaofei Li

Response to Comments on sustainability-2864453: PM2.5 and O3 in an enclosed basin, Guanzhong Basin of Northern China: Insights into distributions, appointment sources, and transport pathways by Xiaofei Li et al.

We thank the reviewer 1 for their constructive and thoughtful comments on our manuscript.

Editor and Reviewer comments:

Thank you for your effort to revise the manuscript.

The manuscript revised according to the comments of reviewer. The manuscript has been improved much better.

But biomass burning factor is a big problem in my recognition. I have checked the reference you mentioned. In Li et al (2023), they mentioned that "A factor with the relatively higher load of K+, Na+, and Cl− is defined as BOB." So, your argument of high loading of EC and OC cannot stand.

From my knowledge, you can check the OC/EC (60.3) and Char-EC/Soot-EC (>10), to see if they are from BOB or not. More information is from the following paper: Characteristics and sources of the fine carbonaceous aerosols in Haikou, China - ScienceDirect

Your results showed that BOB occupied for 26-38% of PM2.5, which is a very large source. The sampling period was in summer, so it's very low chance there's a large source of BOB. There should be more evidence to support your classification such as hotspot during sampling time.

So, my advice is to check OC/EC and Char-EC/Soot-EC in your data, and hotspot during your sampling period to confirm factor 3. Otherwise, you can name factor 3 as Others.

Response: Thank you very much for your comments, and we are also aware of this mistake. In response to this question, we referred to a large number of literature and rerun the PMF model, the PMF model results show that there is still a high loading of OC and EC. We also referred to the reviewer's suggestions and used OC/EC values to determine the source of pollutants. In summer, OC/EC values were 9.84 ± 6.95 and 10.00 ± 6.99 in XN and QL, respectively. Some studies have shown that when the OC/EC value is between 2.5 and 10.5, it represents a coal combustion source (COB)[1], so it is determined as COB based on the OC/EC values. So factor 3 was defined as COB due to the high loading of EC and OC. In XN and QL, the contribution rate of COB is 38% and 26%, respectively. Referring to the contribution rate of COB in Haikou during summer (23.8%)[2], I think that factor 3 is a reasonable explanation for COB in XN and QL.

Some minor comments:

Line 140: 2.4.248. h back trajectory ->2.4.2. 48-hour backward trajectory

Response: Done

Line 344 3.3.248. h back trajectory

Response: Done

Line 440-441: do the authors mean the PM2.5 in XN was lower than QL in winter?

Response: Thank you for your suggestion. We have made the correction. The meaning of this sentence is thatThe concentration of PM2.5 in XN (53.40 ± 17.42 μg m−3) was higher than that in QL (27.57 ± 8.27 μg m−3), but the concentrations of PM2.5 in XN and QL in summer were significantly lower than in winter.”

References

[1] Chen Y J, Zhi G R, Feng Y L, et al. Measurements of emission factors for primary carbonaceous particles from residential raw-coal combustion in China[J]. Geophysical Research Letters, 2006, 33( 20), doi: 10. 1029 /2006GL026966.

[2] Liu B, Zhang J, Wang L, et al. Characteristics and sources of the fine carbonaceous aerosols in Haikou, China[J]. Atmospheric Research, 2018, 199: 103-112.

Reviewer 4 Report

Comments and Suggestions for Authors

Enhancements have been made to the content and presentation of the paper.

However, it is essential to

acknowledge that despite these advancements, the paper still carries certain limitations that warrant attention. To ensure transparency and clarity, the authers should incorporate a dedicated subsection in the main manuscript titled "Limitations." In this subsection, they  can meticulously outline all identified limitations, including but not limited to the restricted number of chemical speciation included in the Positive Matrix Factorization (PMF) modeling.

By explicitly stating these limitations,  readers will be provided with a comprehensive understanding of the research scope and the factors that may influence the interpretation of its findings.

Comments on the Quality of English Language

Moderate English editing is required.

Author Response

Dear Editor:

  We highly appreciate your hard work and positive comments to improve our MS. We have tried our best to improve the manuscript in the current revised version according to the suggestions from the reviewer 4.

  The response to reviewer 4 and the major modifications in the manuscript have been marked in red color.

  If you have further questions or suggestions, please let me know.

                                             Yours Xiaofei Li

Response to Comments on sustainability-2864453: PM2.5 and O3 in an enclosed basin, Guanzhong Basin of Northern China: Insights into distributions, appointment sources, and transport pathways by Xiaofei Li et al.

We thank the reviewer 4 for their constructive and thoughtful comments on our manuscript.

Editor and Reviewer comments:

Enhancements have been made to the content and presentation of the paper.

However, it is essential to acknowledge that despite these advancements, the paper still carries certain limitations that warrant attention. To ensure transparency and clarity, the authers should incorporate a dedicated subsection in the main manuscript titled "Limitations." In this subsection, they can meticulously outline all identified limitations, including but not limited to the restricted number of chemical speciation included in the Positive Matrix Factorization (PMF) modeling.

By explicitly stating these limitations, readers will be provided with a comprehensive understanding of the research scope and the factors that may influence the interpretation of its findings.

Response: Thank you for your suggestion. We have added 3.4 Uncertainties:

“When using the PMF model for pollution source analysis, there should be no less than 20 species in principle[1]. In this study, only PM2.5, OC, EC and water-soluble ions were considered, which may lead to incomplete source analysis results. On the other hand, the PMF model is not suitable for substances with short lifespan, strong activity, or a large number of secondary sources, which may lead to an underestimation of the relative contribution of natural and transportation sources. But during the model operation process, we strictly adhere to the uncertainty calculation basis, which is 5% of the measured value plus the minimum detection limit. In future research, we will also consider inputting metal elements to obtain more comprehensive source apportionment results.

There are some uncertainties in the air mass trajectory model, which mainly include[2-3]:(1) Numerical methods for model calculations. (2) Wind field error. (3) Position error, difference between model terrain and real terrain. (4) The chaotic nature of the atmosphere makes it difficult to accurately calculate trajectories. In this study, we only relied on the longitude and latitude, date, time of observation points, and meteorological data provided by GDAS, without considering the emission area of pollution sources and estimated emissions. In recent years, the accuracy of trajectory calculation has been improved, especially in meteorological data obtained from weather forecast numerical models. Therefore, in this study, we rely on this qualitative method, which can still determine the source of pollutants. However, in the future, more refined particle dispersion models are needed to obtain quantitative analysis.”

References

[1] Reff A, Eberly S I, Bhave P V. Receptor modeling of ambient particulate matter data using positive matrix factorization: Review of existing methods[J]. Journal of the Air and Waste Management Association, 2007, 57(2): 146-154

[2] Stohl A. Computation, accuracy and applications of trajectories–A review and bibliography. Atmospheric Environment, 1998, 32(6), 947-966.

[3] Li X, Kang S, Sprenger M, et al. Black carbon and mineral dust on two glaciers on the central Tibetan Plateau: sources and implications[J]. Journal of Glaciology, 2020, 66(256): 248-258.

Round 3

Reviewer 1 Report

Comments and Suggestions for Authors

Thank you for your effort on revising the manuscript. 

I agreed with your explanation on factor 3. 

Reviewer 4 Report

Comments and Suggestions for Authors

The authors have fully addressed the suggested comments.

Comments on the Quality of English Language

Minor English editing is required.